# Bridging Quantitative Optimization and Qualitative Reasoning: LLM-Enhanced Neural Architecture Search with Synergistic Weights

## Abstract

Differentiable Neural Architecture Search (NAS) has revolutionized deep learning by automating architecture design, but still faces two interdependent limitations: unreliable connection evaluation based solely on local edge weights leads to suboptimal architecture discretization, while static search spaces prevent the discovery of innovative patterns for optimization. Existing approaches treat these as separable problems, lacking either architectural insight or quantitative grounding. To bridge the gap between quantitative optimization and qualitative reasoning directly, we propose SWNAS, which introduces two key innovations: (1) Synergistic Weights that combine edge and node importance for globally-aware architecture evaluation, overcoming myopic local optimization limitations, and (2) Large Language Model (LLM)-guided dynamic search space evolution that enables intelligent topology expansion beyond fixed constraints. Unlike indirect code generation or heuristic rules, SWNAS directly reason with quantitative structural signals to refine discretization and guide strategic node placement, establishing true large-small model collaboration. Extensive experiments demonstrate SWNAS's effectiveness: achieving 2.33% error rate on CIFAR-10 and 23.9% on ImageNet, while maintaining computational efficiency. Our modular design enables seamless integration into existing DARTS-family methods, consistently improving performance by 0.16-0.19% across frameworks. Importantly, SWNAS demonstrates robust generalizability across different search spaces and maintains stable performance across multiple LLMs, demonstrating that genuine quantitative-qualitative integration can systematically advance neural architecture discovery.

## 1 Introduction

Neural Architecture Search (NAS) has transformed deep learning by automating the traditionally manual process of architecture design Pham et al. (2018); Liu et al. (2018). Among the diverse NAS approaches, Differentiable Architecture Search (DARTS) Liu et al. (2019) represents a significant breakthrough, reducing computational costs from thousands of GPU days required by reinforcement learning methods Zoph & Le (2017) and evolutionary algorithms Real et al. (2019) to mere single-digit GPU days through gradient-based optimization.

The critical challenge in differentiable NAS is the gap between quantitative optimization and qualitative reasoning that effective architectural design requires. This gap manifests in two interdependent critical limitations that severely constrain the potential of existing methods.

First, DARTS series methods often suffer from myopic optimization, primarily due to their reliance on locally computed edge-level weights for connection evaluation. A key underlying issue is the significant variation in output feature map scales across different operations and nodes in the network. While architecture weights in DARTS reflect the relative importance of operations locally (within the same edge or from the same node), they lack global comparability because the magnitude of a weight's influence depends on the scale of its corresponding input feature map (source node). Consequently, directly comparing these raw weights leads to suboptimal architectures.

Second, existing approaches are fundamentally constrained by fixed discretization rules and search space boundaries that prevent the discovery of innovative architectural patterns. While recent efforts

have attempted progressive expansion Chen et al. (2021) and partial connections Xu et al. (2019), they still operate within predetermined supernet boundaries, and their discretization is typically enforced through fixed discretization rules (e.g. DARTS chooses top-2 edges per node), which further restricts the emergence of unconventional yet potentially effective architectures.

Current attempts to address these limitations through attention mechanisms Sun et al. (2022) and dynamic search strategies Ci et al. (2021) remain limited by their indirect optimization paradigm. Even recent explorations leveraging Large Language Models (LLMs) for architecture generation Nasir et al. (2024) or design principle extraction Zhou et al. (2025) operate through indirect mechanisms—code generation or external performance feedback—rather than achieving deep integration of quantitative optimization with qualitative architectural reasoning.

To address these challenges, we propose SWNAS, a groundbreaking framework that pioneers the deep integration of weight-based optimization with LLM-driven reasoning to bridge the critical gap between quantitative optimization and qualitative reasoning. Two revolutionary innovations are introduced: (1) **Synergistic Weight** that combines edge and node importance for globally-aware architecture evaluation, overcoming myopic local optimization limitations; and (2) **LLM-guided dynamic search space evolution** that enables intelligent topologies that transcend fixed constraints. Given the inherent complexity of synergistic weights and global topology, simple heuristic rules fall short in ensuring structural completeness and balance. Therefore, our approach leverages the reasoning capabilities of LLMs to refine architecture discretization and guide node expansion, deriving principled architectural evolution.

Our contributions are summarized as follows:

- **Synergistic Weights for Globally-Aware Architecture Search**: We introduce Synergistic Weights that integrate edge and node importance measures, overcoming prior local optimization limitations and providing a more robust, holistic assessment of information flow patterns that captures global architectural behavior beyond myopic edge-level optimization.

- **LLM-Guided Topology Refinement via Synergistic Weight**: We pioneer the use of LLMs as high-level reasoning engines that directly interpret quantitative synergistic weight data with structural information to enable qualitative discretization refinement and dynamic node placement strategies. This novel synergy bridges the optimization-reasoning gap, enabling adaptive space expansion and sophisticated architecture discovery unattainable by traditional quantitative-only methods, transcending static search space boundaries.

- **State-of-the-Art Performance with Modular Integration**: We achieve remarkable effectiveness with 2.33% error rate on CIFAR-10 and competitive ImageNet performance (23.9% top-1 error) while maintaining computational efficiency (0.16 GPU days). Our modular design enables seamless integration into existing DARTS-family frameworks, consistently improving performance by 0.16-0.19 percentage points across different methods and search space, demonstrating the practical value of synergistic optimization-reasoning integration for enhancing current NAS frameworks.

## 2 RELATED WORKS

The rapid advancement of NAS has broadened its applications across diverse domains, including facial expression recognition Li et al. (2021b) and brain signal analysis Li et al. (2021a; 2022). However, NAS still faces the fundamental challenge of bridging quantitative optimization and qualitative reasoning that effective architectural design requires. Recent research has addressed this gap through three main approaches.

### 2.1 OPTIMIZATION-BASED SOLUTIONS FOR CONNECTION EVALUATION

Current differentiable NAS methods attempt to address unreliable edge weight comparisons through refined optimization strategies. SharpDARTS Hundt et al. (2019) improved learning rates and regularization, DARTS+ Liang et al. (2019) used early stopping to limit problematic skip connections, FairDARTS Chu et al. (2020b) recalibrated operation weights to reduce bias and $\beta$-DARTS Ye et al. (2022) introduced beta-decay regularization. These approaches remain fundamentally limited by lo-

cal optimization paradigms, focusing on individual edge characteristics rather than capturing global architectural reasoning about information flow patterns.

## 2.2 ATTENTION-ENHANCED ARCHITECTURE ASSESSMENT

Attention mechanisms have been incorporated to improve connection importance evaluation beyond raw weight comparisons. AGNAS Sun et al. (2022) employed attention modules for adaptive weight adjustment, while SE Hu et al. (2018) and CBAM Woo et al. (2018) introduced channel and spatial attention mechanisms for enhanced feature recalibration. However, these methods still focus primarily on edge-level or channel-level importance without integrating global node-level reasoning about architectural topology.

## 2.3 SEARCH SPACE EVOLUTION STRATEGIES

Breaking topology limitations represents a critical challenge in NAS, as fixed search spaces constrain architectural innovation to pre-conceived patterns. Several approaches have attempted to introduce dynamism into search space design. PDARTS Chen et al. (2021) addressed architecture discrepancy through progressive network deepening, gradually expanding cell complexity during the search process. PC-DARTS Xu et al. (2019) developed partial channel connections to maintain flexibility while reducing memory consumption. NSE Ci et al. (2021) proposed evolving search space subsets with dynamic refilling mechanisms, allowing selective exploration of architectural components.

These methods fundamentally remain constrained by their reliance on predetermined supernet topologies and rule-based expansion strategies. They lack the sophisticated reasoning capabilities needed to identify truly innovative architectural patterns that transcend existing design paradigms.

## 2.4 LARGE LANGUAGE MODELS FOR ARCHITECTURE REASONING

Recent work has explored LLMs as external reasoners for architecture design, representing a fundamentally different paradigm from optimization-based approaches. LLMatic Nasir et al. (2024)uses LLMs for architecture mutation based on performance feedback, while LAPT Zhou et al. (2025) extracts design principles for search space pruning. RZ-NAS Ji et al. (2025) employs LLMs with training-free metrics and reflective modules to comprehensively understand the search tasks and architectures from text/code levels, while NADER Yang et al. (2025) formulates neural architecture design as a multi-agent collaboration problem with graph-based representations by learning from past experiences.

**Critical distinction**: These approaches use LLMs as external black-box code generators or optimizers, operating through qualitative performance feedback rather than directly interpreting quantitative architectural data. Our proposed SWNAS fundamentally differs by integrating LLMs deeply to reason about Synergistic Weights and global topology, enabling direct integration of quantitative optimization signals with qualitative architectural reasoning, rather than treating them as separate processes.

## 2.5 THE OPTIMIZATION-REASONING GAP AND OUR APPROACH

Existing NAS methods face a dichotomy: excelling at quantitative gradient-based optimization but lacking global reasoning, or incorporating reasoning heuristics while being disconnected from optimization dynamics. This leads to myopic connection evaluation and static search spaces.

We bridge this gap by unifying quantitative weight optimization with qualitative architectural reasoning. Unlike methods treating optimization and reasoning separately, our approach uses LLMs to reason about Synergistic Weights—combining edge and node importance—to guide both discrete architecture selection and dynamic search space evolution.

## 3 METHODOLOGY

Effectively bridging quantitative optimization and qualitative reasoning in NAS requires accurately assessing connection importance while transcending predefined cell structures. SWNAS introduces

a comprehensive framework that addresses these challenges through a core Synergistic Weighting mechanism and two complementary strategies. Our approach simultaneously evaluates edge importance and node significance, then leverages these synergistic weights to develop an enhanced discretization strategy and adaptively incorporate new computational nodes by identifying critical information flow paths.

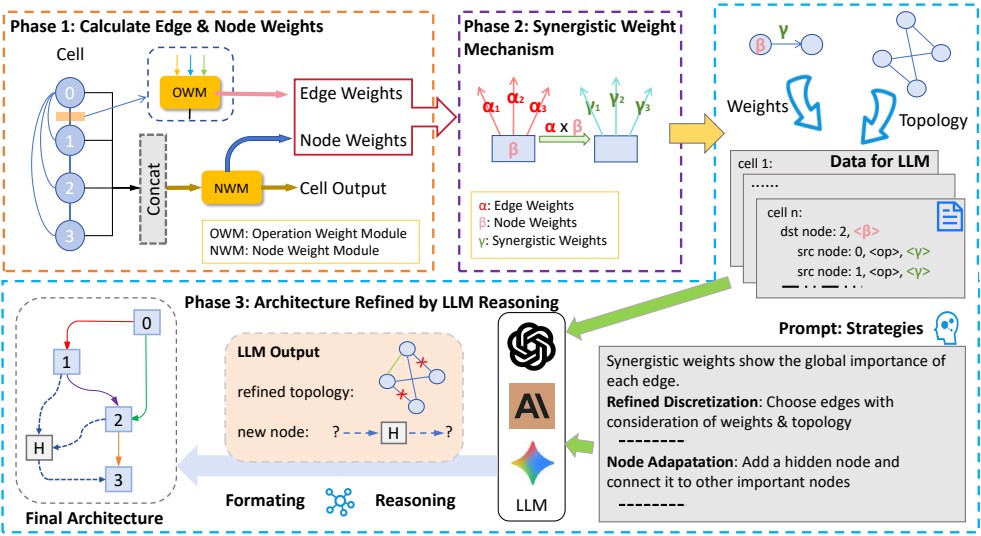

Figure 1: Overall Framework of SWNAS. Phase 1 illustrates the Edge Weights and Node Weights computing process by OWM and NWM. Phase 2 depicts the mechanism of Synergistic Weights. Phase 3 outlines the data structures and prompt designs used for the LLM. It also details how the suggestions generated by the LLM are integrated into the architecture.

### 3.1 OVERALL FRAMEWORK

Figure 1 depicts the overall framework of SWNAS. In Phase 1, edge weights and node weights are computed via the operation Weights Module (OWM) and Node Weights Module (NWM) during the neural architecture search stage. In Phase 2, Synergistic Weights are obtained by integrating each edge weight with the node weight of its corresponding source node. In Phase 3, the resulting weights and architectural topology are organized into a structured format suitable for input into a large language model (LLM). Guided by strategy-specific prompts, the LLM performs reasoning to infer optimized connection patterns and potential node expansions. The final architecture is then refined based on the LLM's recommendations.

### 3.2 SYNERGISTIC WEIGHTS MECHANISM

This section details the mechanism for computing synergistic edge and node weights, which are instrumental in deriving global relative edge importance. Intuitively, an edge's contribution should depend not only on its operation but also on the importance of its source node. Thus, we first derive edge weights and node weights separately.

DARTS formulates the architecture search problem as a continuous optimization task. It assigns learnable weights to all candidate operations on each edge of a cell, enabling gradient-based optimization of both architecture weights and network parameters. Each cell is a DAG where nodes receive 2 input edges from previous cells, and internal edges connect nodes within the cell; node outputs are concatenated to form the cell output. Cells are stacked to build the final network.

Inspired by AGNAS Sun et al. (2022), we utilize attention modules for weight computation. Input features are first concatenated, then processed by Global Average Pooling (GAP). The pooled features pass through a two-Fully Connected (FC) layer bottleneck employing ReLU and Sigmoid

activations to generate channel-wise attention scores. These scores recalibrate the concatenated input features via element-wise multiplication.

Specifically, the OWM evaluates $K$ candidate operations $\{op_1, \ldots, op_K\}$ on each edge and calculates operation weights $\mathcal{A}_k$. The edge weight $\alpha_e$ is then the maximum $\mathcal{A}_k$ (Defined by DARTS):

$$\alpha_e = \max_k(\mathcal{A}_k) \tag{1}$$

Similarly, the NWM assesses the importance of output nodes in the cell, each providing a feature map $X_p \in \mathbb{R}^{B \times C \times H \times W}$. These feature maps are concatenated along the channel axis (axis 1). This results in a single tensor, $X_{\text{concat}}$, with dimensions $\mathbb{R}^{B \times (M \cdot C) \times H \times W}$.

$$X_{\text{concat}} = \text{Concatenate}(X_1, \ldots, X_M) \tag{2}$$

The NWM then generates channel attention scores $W_{\text{ch\_att}} \in \mathbb{R}^{B \times (M \cdot C) \times 1 \times 1}$ using its own FC layers, which re-weights $X_{\text{concat}}$ to produce the cell's output $X_{\text{cell\_out}}$.

$$W_{\text{ch\_att}} = \sigma(\text{FC}_2(\text{ReLU}(\text{FC}_1(\text{GAP}(X_{\text{concat}}))))) \tag{3}$$

$$X_{\text{cell\_out}} = X_{\text{concat}} \odot W_{\text{ch\_att}} \tag{4}$$

Output $X_{\text{cell\_out}}$ of the current cell serves as the input $C_{in}$ to the subsequent cells. In this way, NWM are integrated into the computational graph, enabling optimization via gradient descent.

The node weight $\beta_n$ for a source node $n$ is the sum of the $C$ channel attention values in $W_{\text{ch\_att}}$ corresponding to that node's feature map.

Finally, to form a synergistic weight $\Gamma_e$, we adopt multiplication, as the significance of an edge is jointly determined by both its source node and the edge itself, a dependency that is most appropriately represented through their product.

$$\Gamma_e = \beta_n \cdot \alpha_e \tag{5}$$

The value of $\Gamma_e$ guides the final architecture generation. SWNAS employs a two-stage search algorithm: Stage 1 determines $\alpha_e$ (OWM only) by weighting each edge's local importance; Stage 2 introduces and trains the NWM for $\beta_n$ with its source node's global significance, then calculates the final $\Gamma_e$. This ensures that edges from globally important nodes receive higher priority, preventing the selection of locally optimal but globally suboptimal connections. Algorithm 1 outlines this process.

---

**Algorithm 1** Two-Stage Architecture Search for Collaborative Weights

---

1: **Input:** Network $S_{\text{net}}$, OWM, NWM, $D_{\text{train}}$, $D_{\text{val}}$, $E_{\text{total}}$, $E_{\text{NWM\_start}}(< E_{\text{total}})$.
2: **Output:** $\alpha_e, \beta_n, \Gamma_e$.
3: **Stage 1: Operation Weight Search (Epochs 1 to $E_{\text{NWM\_start}} - 1$)** // NWM delayed introduction to prevent training instability and ensure sufficient supernet convergence
4: Construct $S_{\text{net}}$ with all candidate operations and OWM on each edge.
5: **for** epoch $e = 1$ to $E_{\text{NWM\_start}} - 1$ **do**
6:     Update network and OWM parameters $w, \theta_{\text{OWM}}$ on $D_{\text{train}}$: $\nabla_{w, \theta_{\text{OWM}}} \mathcal{L}_{\text{train}}(w, \theta_{\text{OWM}})$.
7: **end for**
8: **Stage 2: Node Weight Search (Epochs $E_{\text{NWM\_start}}$ to $E_{\text{total}}$)**
9: Integrate NWM at cell output aggregation points in $S_{\text{net}}$.
10: **for** epoch $e = E_{\text{NWM\_start}}$ to $E_{\text{total}}$ **do**
11:     Update $w, \theta_{\text{OWM}}, \theta_{\text{NWM}}$ on $D_{\text{train}}$: $\nabla_{w, \theta_{\text{OWM}}, \theta_{\text{NWM}}} \mathcal{L}_{\text{train}}(w, \theta_{\text{OWM}}, \theta_{\text{NWM}})$.
12: **end for**
13: On $D_{\text{val}}$: Compute all $\mathcal{A}_{(\text{edge,op})}$ using learned OWM and all $\beta_n$ using learned NWM.
14: For each edge: $\alpha_e = \max_{\text{op}}(\mathcal{A}_{(\text{edge,op})})$; calculate $\Gamma_e = \beta_n \cdot \alpha_e$.
15: **Return** All $\alpha_e, \beta_n, \Gamma_e$.

---

## 3.3 REFINED DISCRETIZATION STRATEGY

Synergistic weights provide a global measure of importance for each candidate edge within the cell. However, simply selecting the top-k edges with the highest $\Gamma_e$ often leads to topologically imbalanced networks. Specifically, the cell width—number of input edges to each intermediate node—is a critical determinant of performance. A naive top-k selection fails to ensure healthy cell width, often resulting in overly deep and narrow pathways that limit expressive power.

Accordingly, SWNAS introduces an LLM-driven discretization strategy accounting for both weight prioritization and topology. The LLM first establishes initial input connections based on node importance ($\beta_n$), directing information to the most critical nodes. It then selects

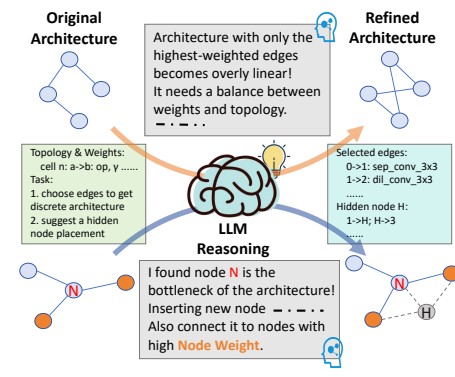

Figure 2: LLM Reasoning Strategies

internal edges in descending $\Gamma_e$ order until the budget is met. Beyond local selection, the LLM conducts global topological reasoning to ensure integrity of the network, producing a discretized architecture with competitive performance and coherent topology.

## 3.4 NODE ADAPTATION STRATEGY

After initial discretization, SWNAS employs an LLM-guided Node Adaptation Strategy to further enhance representational capacity. While the discretized cell provides a strong baseline, it may still have bottlenecks or lack complexity. This strategy dynamically expands the cell architecture.

The LLM analyzes the complete cell topology—including connection patterns and node importance ($\beta_n$)—to identify strategic locations for new "hidden nodes" (not serving as outputs). It identifies critical nodes or overly linear structures within the DAG by analyzing its pathways, then introduces new nodes to ensure non-trivial topological complexity.

Once proposed, the augmented search space undergoes a brief targeted training to finalize connections. This two-stage process (Algorithm 2)—initial discretization followed by adaptive node expansion—embodies a coarse-to-fine search, efficiently evolving the network to a high-performance, structurally sophisticated final architecture.

---

**Algorithm 2** LLM-Guided Two-Stage Architecture Discretization and Node Adaptation

---

1: **Input:** Search space $S$; $\Gamma_e$; $\beta_n$; LLM; Prompts $P_1$, $P_2$ (Discretization & Adaptation); Training dataset $D_{\text{train}}$; Adaptation training epochs $E_{\text{adapt}}$.
2: **Output:** Final discretized cell architecture $A_{\text{final}}$.

3: **Stage 1: LLM-Driven Refined Discretization**
4: Organizing $\Gamma_e$, $\beta_n$ and topology of $S$ into a structured format $F_1$.
5: $S \leftarrow \text{Apply}(\text{Format} \leftarrow \text{LLM}(P_1, F_1))$ {query LLM and apply discretization to search space}

6: **Stage 2: LLM-Guided Node Adaptation**
7: Organizing $\Gamma_e$, $\beta_n$ and topology of discretized $S$ into a structured format $F_2$.
8: $S \leftarrow \text{Apply}(\text{Format} \leftarrow \text{LLM}(P_2, F_2))$
9: Initialize the new edges with full operations and OWM. {The new search space $S$ does not integrate NWM}
10: **for** epoch $e = 1$ to $E_{\text{adapt}}$ **do**
11:     Update weights of $A_{\text{search}}$ using $D_{\text{train}}$.
12: **end for**
13: $A_{\text{discrete}} \leftarrow \text{discrete}(S)$ {Select operations with highest weighs of each edge}

14: **Return** $A_{\text{discrete}}$.

---

## 4 EXPERIMENTS

### 4.1 DATASETS AND SETTINGS

CIFAR-10 Krizhevsky et al. (2009) contains 60,000 32×32 color images, split into 50,000 for training and 10,000 for testing. During search, the training set is split equally into search and validation subsets. For transferability, the discovered architecture is applied to ImageNet Krizhevsky et al. (2017) with 1.28M training and 50K validation images across 1,000 classes.

Our search space targets normal and reduction cells in a directed acyclic graph, each receiving inputs from two preceding cells, containing four initial intermediate nodes, and aggregating outputs from them. Edges may implement one of eight operations: none, skip connect, avg/max pool 3×3, sep conv 3×3/5×5, or dil conv 3×3/5×5. Cells are searched for general positions and stacked to form deeper models. The discrete network includes 8-10 edges, 1 hidden node and 2 new edges per cell.

The search runs in phases: initial architecture search for 50 epochs with cell depth 8, activating node attention after epoch 45. Both search and node expansion use batch size 64, cross-entropy loss, and SGD (initial LR 0.025, cosine decay to 0.001, momentum 0.9, weight decay $3 \times 10^{-4}$). Node expansion trains for 30 epochs under the same settings. All searches use an L40s GPU (48 GB). We used Gemini-2.5-pro (Cloud API, without fine-tuning and prior knowledge) for reasoning.

### 4.2 RESULTS ON CIFAR-10 AND IMAGENET

Table 1: Comparison of Test Error Rate(%), Params(M) and Cost(GPU days) on CIFAR-10

| Algorithm | Error | Params | Cost |
|---|---|---|---|
| NASNet-A Zoph et al. (2018) | 2.65 | 3.3 | 1800 |
| ENAS Pham et al. (2018) | 2.89 | 4.6 | 1 |
| AmoebaNet-B Real et al. (2019) | 2.55 | 2.8 | 3150 |
| DARTS(2nd) Liu et al. (2019) | 2.76 | 3.3 | 4.0 |
| DARTS- Chu et al. (2020a) | 2.50 | 3.5 | 0.4 |
| PDARTS Chen et al. (2021) | 2.50 | 3.4 | 0.3 |
| PC-DARTS Xu et al. (2019) | 2.57 | 3.6 | 0.1 |
| Fair DARTS Chu et al. (2020b) | 2.54 | 2.8 | 0.4 |
| AGNAS Sun et al. (2022) | 2.53 | 3.6 | 0.4 |
| $\beta$-DARTS Ye et al. (2022) | 2.53 | 3.75 | 0.4 |
| U-DARTS Huang et al. (2023) | 2.59 | 3.3 | 3.0 |
| SWD Xue et al. (2024) | 2.51 | 3.17 | 0.13 |
| IS-DARTS He et al. (2024) | 2.56 | 4.25 | 0.42 |
| LLMatic Nasir et al. (2024) | 5.74 | - | - |
| LAPT-REA Zhou et al. (2025) | 2.65 | - | 0.1 |
| SWNAS | **2.33** | 4.1 | 0.16 |

Table 2: Comparison of Test Error Rate(%), Params(M) and Search Cost(GPU days) on ImageNet

| Algorithm | Top-1 | Top-5 | Params | Cost |
|---|---|---|---|---|
| NASNet-A Zoph et al. (2018) | 26.0 | 8.4 | 5.3 | 1800 |
| AmoebaNet-A Real et al. (2019) | 25.5 | 8.0 | 5.1 | 3150 |
| DARTS(2nd) Liu et al. (2019) | 26.9 | 9.0 | 4.9 | 4.0 |
| PDARTS Chen et al. (2021) | 24.7 | 7.5 | 5.1 | 0.3 |
| PC-DARTS Xu et al. (2019) | 25.1 | 7.8 | 5.3 | 0.1 |
| RF-DARTS Zhang et al. (2023) | 24.0 | 7.2 | - | - |
| IS-DARTS He et al. (2024) | 24.1 | 7.1 | 6.4 | 0.42 |
| SWD Xue et al. (2024) | 24.5 | 7.6 | 6.3 | 0.13 |
| DARTS-† Chu et al. (2020a) | 23.8 | 7 | 4.9 | 4.5 |
| AGNAS† Sun et al. (2022) | 25.4 | 7.9 | 6.7 | 3.3 |
| LAPT-REA Zhou et al. (2025) | 24.9 | - | - | 2.0 |
| SWNAS | 23.9 | 7 | 7.1 | 0.16 |

† directly searched on ImageNet. Others searched on CIFAR-10 then transferred (including SWNAS).

To evaluate the capability of searched cells, a 20-cell network was developed and trained. The training process employed an optimizer with a batch size of 96 and 600 epochs. Network parameters were refined via the SGD algorithm, incorporating a momentum of 0.9 and a weight decay of $3 \times 10^{-4}$. The learning rate was initially set to 0.025 and subsequently reduced to zero using a cosine annealing approach. Further, a cutout length of 16 was implemented, the auxiliary loss weight was established at 0.4, and the drop path probability was designated as 0.2. The results in Table 1 show that our approach achieved 97.67% accuracy on CIFAR-10, exceeding the results of many state-of-the-art methods. The search cost includes the normal search phase, node adaptation phase (0.12 + 0.03 GPU days) and LLM reasoning process (0.01 GPU days), maintaining computational efficiency. The LLM query costs about 6000 input and 10000 output tokens for each run.

To verify the transferability and robustness of our methods, we used the architecture searched on CIFAR-10 to retrain on ImageNet. We trained the network for 250 epochs. The batch size was 256, and the network had 20 layers. We used SGD to update the parameters. The momentum was 0.9, and the weight decay was $3 \times 10^{-5}$. We started with a learning rate of 0.1. We used a cosine annealing strategy to lower the learning rate to 0. Also, the cutout length was 16. The drop path probability was 0.2, and the auxiliary loss weight was 0.4. As in Table 2, the architecture we found by SWNAS on CIFAR-10 got 76.1% top-one and 93% top-five test accuracy when transferred to ImageNet, which is better than all transferred algorithms.

## 4.3 ABLATION STUDY

### SYNERGISTIC WEIGHTS: TRANSCENDING LOCAL OPTIMA

The effectiveness of our Synergistic Weighting scheme is evidenced by the substantial performance improvements. We measured the correlation between Synergistic Weights and accuracy improvement of internal edges on CIFAR-10 by fixing a cell's input edges and benchmarking the performance with six internal edges separately. Figure 3 demonstrates that edges ranked higher by our method contribute significantly more to network performance, with a steeper positive slope in accuracy compared to raw weight rankings. This validates our core synergistic mechanism provides a more holistic assessment of information flow patterns.

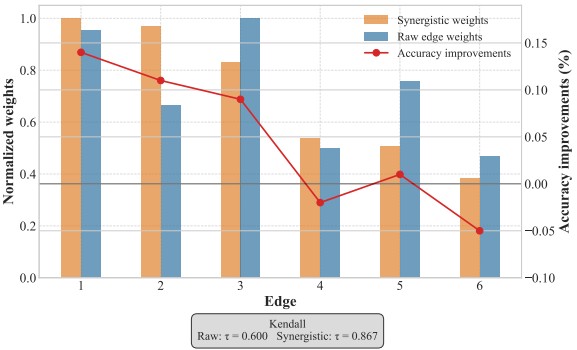

Figure 3: Correlation between Raw/Synergistic Weights and Acc Improvements of 6 internal edges

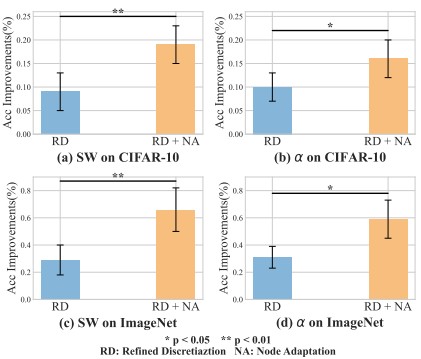

Figure 4: Accuracy Improvements of SWNAS Modules over Baseline

### REFINED DISCRETIZATION AND NODE ADAPTATION: BEYOND FIXED TOPOLOGY

The node attention module and node expansion mechanism in SWNAS feature a plug-and-play design, enabling seamless integration into most DARTS-family methods. To evaluate this modularity and the improvements from LLM reasoning guided Discretization Refinement and Node Adaptation, we conducted five independent runs on CIFAR-10. To validate cross-framework compatibility, we replaced OWM with traditional DARTS $\alpha$ parameters (keep NWM) and conducted another 5 runs. In each run, we compared three configurations:

- Baseline: Synergistic Weights without Refined Discretization and Node Adaptation, using traditional rules to discretize (top-2 edges per node).

- RD: Synergistic Weights and Discretization Refinement without Node Adaptation.

- RD + NA: A network utilizing full SWNAS framework.

Results in Figure 4 demonstrate that SWNAS modules provide consistent improvements even when integrated with standard DARTS components, confirming their modular nature. This compatibility highlights a key advantage: SWNAS components can be adopted independently as drop-in enhancements for existing NAS frameworks without requiring architectural overhaul.

### UNIVERSALITY AND TRANSFERABILITY ON DIFFERENT SEARCH SPACE

To validate the broad applicability of SWNAS in the field of NAS beyond DARTS, we tested SWNAS on NAS-Bench-201. We also integrated SWNAS into other DARTS-series to verify the transferability (supplementary materials). These experiments consistently demonstrate that SWNAS exhibits stable superior generalizability, search efficiency, and robustness across diverse search spaces.

## 5 DISCUSSIONS

### 5.1 QUALITATIVE REASONING VIA DIFFERENT STRATEGIES AND LLMS

Due to the stochastic nature of the qualitative reasoning process, the outcomes can be opaque without reliable strategies. In our approach, this uncertainty is reflected in the global topological control during the process of Refined Discretization and Node Adaptation. To make it more controllable and interpretable, we investigate the impact of different strategies on the outcomes.

For a searched cell, we fixed internal edges and varied the cell width, which is the number of input edges (selected by edge weights ranking). As shown in Figure 5, higher accuracy values are concentrated around a cell width of 5–7, allowing the cell to receive sufficient informational input without being overwhelmed by excessive complexity.

In the Node Adaptation phase, we applied three strategies on the same network during the Node Adaptation phase: randomly inserting edges (random operations) to the cell; linking the new node with nodes of low weights/high weights. We conducted the experiments for 5 rounds and tested the average accuracy improvements on CIFAR-10. Table 3 shows that the random strategy unexpectedly resulted in a negative improvement, which confirms that the enhancement brought by our node expansion is not merely due to the increase in model parameters, but rather stems from the strategy-guided LLM reasoning. Furthermore, the experiments demonstrate that connecting new nodes to high-weight nodes yields better performance than connecting to low-weight nodes.

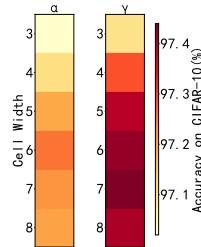

Figure 5: Varying Cell Width on Raw/Synergistic Weights

| Strategy | ΔAcc(%) |
|----------|---------|
| Random | -0.03 |
| Low Weights* | 0.09 |
| High Weights* | 0.13 |

\* Linking the new node with nodes of low/high weights.

Table 3: Comparison of Node Adaptation Strategies

| LLM | Accuracy(%) |
|-----|-------------|
| DeepSeek-R1 | 97.60 |
| Claude-3.7-Sonnet | 97.65 |
| Gemini-2.5-Pro | 97.64 |
| GPT-5 | 97.62 |

Table 4: Performance of Different Large Lauguage Models

These experimental results collectively form the complete strategy of our approach. Building on this strategy, we further investigated the performance of different LLMs in the reasoning process. As shown in Table 4, various LLMs consistently achieve stable performance under our proposed strategy, demonstrating that our qualitative reasoning process is both controllable and interpretable.

### 5.2 COMPUTATIONAL EFFICIENCY ANALYSIS

Despite incorporating an extra search phase, SWNAS maintains computational efficiency comparable to fast differentiable methods. This efficiency comes from strategic and modular design: node expansion is performed in parameter-reduced sub-networks. The added phase requires only 1/6 of the initial search time, introducing minimal overhead for substantial gains.

## 6 CONCLUSION

We have presented SWNAS, a groundbreaking framework that pioneers deep integration of weight-based optimization with LLM-driven reasoning in NAS, bridging quantitative optimization and qualitative reasoning. Its two innovations—Synergistic Weights for globally-aware evaluation and LLM-guided dynamic search space evolution—overcome key limitations of differentiable NAS. Future work includes scaling to larger search spaces, extending beyond vision, and refining LLM reasoning strategies. The modular framework supports these extensions and integration with new NAS paradigms.

ETHICS STATEMENT

We have read and adhere to the ICLR Code of Ethics. We confirm that this submission raises no ethical concerns.

REPRODUCIBILITY STATEMENT

Our codes are available at https://anonymous.4open.science/r/SWNAS-F2D2. We have also attached the cell structure searched on CIFAR-10 in the Appendix (J).

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
