# Supplementary Materials: Bridging Quantitative Optimization and Qualitative Reasoning: LLM-Enhanced Neural Architecture Search with Synergistic Weights

## 1 The Use of Large Language Models

Large language models (LLMs) are integrated into our proposed framework as a core methodological component. Besides, we used LLMs to polish writing. This involved using LLMs to refine sentence structure, improve clarity and coherence, ensure stylistic consistency, and polish the academic tone across the manuscript.

## 2 Preliminaries

Differentiable Architecture Search (DARTS) makes architecture search a continuous optimization problem, speeding it up. DARTS searches within a "cell," a sequence of nodes (feature maps) linked by edges (candidate operations like convolution or pooling).

DARTS relaxes discrete operation choices. For an edge from node $i$ to $j$, its output $O^{(i,j)}(x)$ is a weighted sum of all candidate operations $o \in \mathcal{O}$:

$$O^{(i,j)}(x) = \sum_{o \in \mathcal{O}} \frac{\exp(\alpha_o^{(i,j)})}{\sum_{o' \in \mathcal{O}} \exp(\alpha_{o'}^{(i,j)})} o(x^{(i)}) \tag{1}$$

Here, $x^{(i)}$ is node $i$'s output, and $\alpha_o^{(i,j)}$ are learnable architecture parameters for operation $o$ on edge $(i, j)$, indicating operation importance.

The search optimizes these $\alpha$ parameters. DARTS uses bilevel optimization, alternating between optimizing network weights $\omega$ and architecture parameters $\alpha$:

$$\min_{\alpha} \quad \mathcal{L}_{\text{val}}(\omega^*(\alpha), \alpha) \tag{2}$$

$$\text{s.t.} \quad \omega^*(\alpha) = \arg\min_{\omega} \mathcal{L}_{\text{train}}(\omega, \alpha) \tag{3}$$

$\mathcal{L}_{\text{train}}$ and $\mathcal{L}_{\text{val}}$ are training and validation losses. $\alpha$ are upper-level variables, $\omega$ are lower-level.

**Operation Selection:** After search, a discrete architecture is extracted. For each edge $(i, j)$, the operation $o^*$ with the highest $\alpha_o^{(i,j)}$ is chosen (excluding Zero).

$$o^{(i,j)} = \arg\max_{o \in \mathcal{O}} \alpha_o^{(i,j)} \tag{4}$$

**Edge Selection:** For each intermediate node $j$, the top $k$ (typically $k = 2$) incoming edges $(i, j)$ are selected based on their strongest operation's $\alpha$ (non-Zero). Let $S_j$ be the indices of the top $k$ predecessor nodes for node $j$:

$$S_j = \text{TopK}_{i<j} \left( \max_{o \in \mathcal{O}, o \neq \text{Zero}} \alpha_o^{(i,j)} \right) \tag{5}$$

Retained connections are $\{(i, o^{(i,j)}) | i \in S_j\}$.

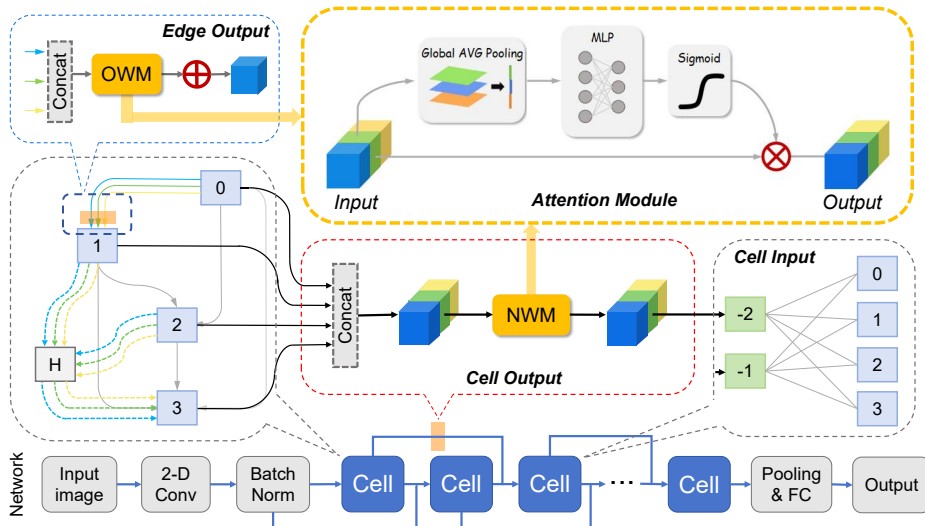

Figure 1: Framework of SWNAS Search Space

Attention-based methods like AGNAS use attention modules for operation weights. These attention weights $\beta$ are optimized with network weights $\omega$ via standard training:

$$\min_{\omega,\beta} \mathcal{L}_{\text{train}}(\omega, \beta) \tag{6}$$

These methods adaptively focus on key feature-level information. Inspired by this, our work uses attention-like ideas in SWNAS to evaluate edge and node importance, aiming to overcome DARTS's connection selection and fixed topology limits for an evolutionary search space.

## 3 DETAILED STRUCTURE OF SWNAS SEARCH SPACE

Figure 1 depicts the search space of SWNAS. The "edge output" diagram (upper-left) shows the edge weight calculation: multiple operation outputs are concatenated and processed through an Operation Attention Module (OWM), with the resulting weighted outputs subsequently split and summed to produce the final edge output. The "cell output" diagram (bottom-center) illustrates node weight calculation: cell nodes are concatenated and fed into a Node Attention Module (NWM), generating weighted outputs that serve as inputs for the subsequent cell (right). OWM and NWM share a similar structure (upper-right). The left cell diagram visualizes the search space evolution mechanism, where a hidden node H establishes connections with existing nodes to expand the cell structure while not directly contributing to the cell output, thereby enhancing architectural flexibility without increasing computational complexity at the output layer.

## 4 DATA STRUCTURE AND PROMPTS FOR LARGE LANGUAGE MODELS

### 4.1 EXAMPLE DATA STRUCTURE FOR LLM

```
Cell 0:
dst_node 0, 0.6324:
    src_node −2: sep_conv_3x3, 0.3190
    src_node −1: max_pool_3x3, 0.2567
dst_node 1, 0.5965:
    src_node −2: dil_conv_3x3, 0.3087
    src_node −1: avg_pool_3x3, 0.2504
    src_node 0: sep_conv_5x5, 0.1968
dst_node 2, 0.6126:
    src_node −2: dil_conv_5x5, 0.3125
```

```
    src_node −1: skip_connect, 0.2689
    src_node 0: sep_conv_5x5, 0.1934
    src_node 1: sep_conv_3x3, 0.1807
dst_node 3, 0.6378:
    src_node −2: dil_conv_3x3, 0.2894
    src_node −1: avg_pool_3x3, 0.2713
    src_node 0: sep_conv_5x5, 0.1893
    src_node 1: dil_conv_3x3, 0.1744
    src_node 2: avg_pool_3x3, 0.1934
```

-2, -1: Inputs of the cell(come from the stem layer of the network or previous cells)

## 4.2 PROMPTS FOR REFINED DISCRETIZATION

"""
The data shows the search result of a DARTS variant for neural network architecture search. I have calculated the feature map output weights for the nodes and the operation weights for each edge. The number after 'dstnode' represents the weight of that node. I have already selected the operation with the highest weight from each edge and calculated the global relative weight of the 6 internal edges. 8 Input edges from -2,-1 do not have node weights, so their weights are raw.

Now, you need to carefully design a sub-network based on this information, selecting 8-10 edges to form the new network. First, consider input nodes -2 and -1 separately. Select 4-7 input edges in total. Input edges originating from different nodes have varying information flow strengths and thus cannot be directly compared. Edge weights originating from the same node are comparable. Select input edges with relatively high and close weights. Second, consider internal connections by ranking the edges based on their global relative weights(weights of interal edges in data are global, they are comparable). Do not try to balance the number of input and internal edges. You should take control of the overall topology. Ensure the cell has enough inputs and the complexity of the inner structure. Additionally, 'skip connect' operations should not form a continuous path, and you should avoid selecting a large number of pooling operations. After identifying candidate edges, you also need to ensure that the architecture effectively forms a Directed Acyclic Graph, and no node is discarded. Please think carefully and provide the result.
"""

## 4.3 PROMPTS FOR NODE EXPANSION

"""
The data pertains to the node feature maps, output node weights for each cell within the DARTS (Differentiable Architecture Search) neural architecture search method, observed after a period of training. You are now required to consider the addition of a hidden node to this cell. This node should not serve as a direct external output of the cell; rather, its purpose is to enhance the structure. It could be connected in parallel with an existing node(overly linear structure), or it could link with a bottleneck node(critical pathways) or nodes with high weight. You must take into account the overall network topology and connectivity to propose a specific solution.
"""

## 5 BRIDGING THE GAP: EMPIRICAL ANALYSIS OF THE QUANTITATIVE-QUALITATIVE BRIDGE

Synergist Weights proposed by our method have successfully achieved globally-aware architecture evaluation of internal connections, overcoming the limitations of myopic local optimization. We also leverage the qualitative reasoning capabilities of large language models (LLMs), which not only optimize internal node connections but also maintain global topological coherence. This integration enables the architecture to achieve both sufficient depth and width, effectively bridging the gap between high-level architectural reasoning and the quantitative optimization provided by synergistic weights.

## 5.1 LLM-Guided Refined Dicretization for Stable Width Control

The discretization strategy proposed by SWNAS serves as a lower bound to maintain sufficient cell width. To investigate whether LLM-based architecture optimization can enforce this principle while preserving strong internal structure, we conducted a series of experiments.

In the experiments, each generated cell underwent two types of architecture discretization: one following the standard DARTS rule (selecting the top-2 weighted edges for each destination node), and the other guided by an LLM using the width-aware strategy. We analyzed the resulting distributions of cell widths across architectures.

As shown in Figure 2, the LLM-driven global topology optimization effectively maintained cell width within the ideal range. This ensured adequate information flow into each cell, leading to more expressive and stable architectures. In contrast, the traditional rule-based discretization produced more extreme cases: too few input edges limited the cell's information intake, while too many input edges oversimplified the internal structure, ultimately degrading performance. These observations align well with our earlier findings on the impact of cell width.

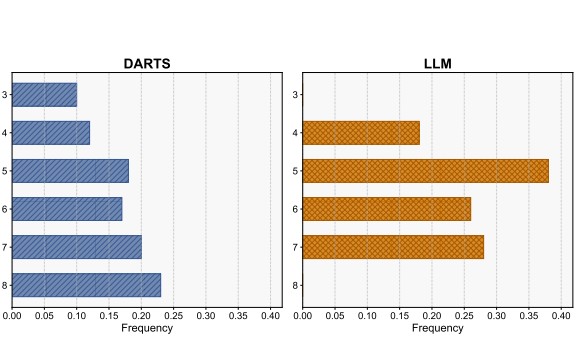

Figure 2: Comparison of Cell Width Distribution Before and After Discretization Refinement

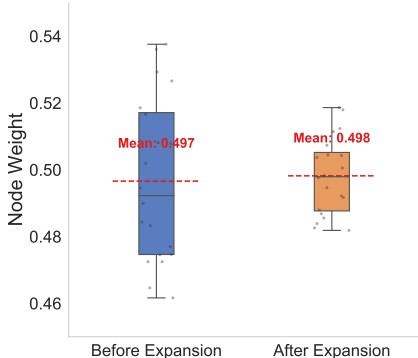

Figure 3: Node Weight Distribution Before and After Node Expansion

## 5.2 Insights of Node Adaptation

The node expansion strategy proposed by SWNAS breaks the limitations of the fixed search space imposed by traditional DARTS-based methods. Instead of relying on a static set of nodes, we introduce a dynamic and adaptive approach to architecture growth. Inspired by heuristic reasoning and leveraging the global structural awareness of large language models (LLMs), our method identifies bottleneck regions or critical information hubs within the current architecture. At these locations, we insert new nodes and connect them to other nodes that exhibit high edge weights, effectively reinforcing the network's ability to capture and distribute essential information.

To gain deeper insight into the mechanism and effectiveness of node expansion, we reintroduced the Node Weights Module (NWM) after the expansion process and recalculated node weights. We then conducted multiple experiments and compared the node weight distributions before and after expansion. As shown in figure 3, the node weights became more concentrated following expansion, indicating a more balanced and effective flow of information across the architecture.

This observation highlights the impact of our node expansion strategy: it not only enhances the expressiveness of the architecture by introducing new processing centers but also alleviates over-reliance on a few dominant nodes. The result is a more resilient and generalizable network structure, capable of better performance.

## 5.3 Synergizing Quantitative Optimization and Qualitative Reasoning

Together, these studies lead to a key insight: relying solely on synergistic weights for quantitative optimization overlooks global topological considerations, potentially resulting in insufficient inputs

or structurally weak designs. Conversely, qualitative reasoning via LLMs promotes coherent global structure but lacks precision in identifying truly critical internal connections.

It is the synergy between the two—quantitative optimization through synergistic weights and high-level topological reasoning via LLMs—that enables robust, high-performing architectures. This dual approach ensures that both the local connection strengths and global design principles are optimized in tandem.

Our empirical findings strongly support this thesis: effective neural architecture search must bridge the gap between fine-grained weight-based optimization and holistic structural reasoning. SWNAS, through its integration of synergistic weights and LLM-guided reasoning, overcomes longstanding limitations in prior NAS approaches—namely, unreliable connection evaluation and static, inflexible search spaces. By fusing data-driven metrics with learned design heuristics, SWNAS delivers architectures that are not only quantitatively strong but also structurally sound, leading to consistent performance gains.

## 6 EXPERIMENTS ON NAS-BENCH-201

To further validate the reliability and generality of our proposed method, we extend the experiments beyond the DARTS search space and evaluate it on NAS-Bench-201 Dong & Yang (2020). NAS-Bench-201 provides a well-defined and exhaustive benchmark covering multiple datasets and a fixed-size search space with complete performance records. This allows us to conduct a broader and more rigorous comparison under controlled conditions.

Due to the restriction of 4 nodes in the NAS-Bench-201 search space, we only applied Synergistic Weights and Discretization Refinement. We keep the hyperparameters the same as they are in the DARTS search space. Although our method is subject to the constraints of the NAS-Bench-201, the results in Table 1 still surpass the majority of existing approaches. This demonstrates not only the effectiveness and robustness of our framework under restricted benchmarks, but also its broader applicability across different NAS spaces, beyond the DARTS setting.

Table 1: Comparison of different NAS methods on NAS-Bench-201.

| Method | CIFAR-10 | | CIFAR-100 | | ImageNet16-120 | |
| --- | --- | --- | --- | --- | --- | --- |
| | Valid Acc (%) | Test Acc (%) | Valid Acc (%) | Test Acc (%) | Valid Acc (%) | Test Acc (%) |
| DARTS(2nd) Liu et al. (2019) | 39.77±0.00 | 54.30±0.00 | 15.03±0.00 | 15.61±0.00 | 16.43±0.00 | 16.32±0.00 |
| PC-DARTS Xu et al. (2019) | 89.96±0.15 | 93.41±0.30 | 67.12±0.39 | 67.48±0.89 | 40.83±0.08 | 41.31±0.22 |
| DARTS- Chu et al. (2020a) | 91.03±0.44 | 93.80±0.40 | 71.36±1.51 | 71.53±1.51 | 44.87±1.46 | 45.12±0.82 |
| AGNAS Sun et al. (2022) | 91.25±0.02 | 94.05±0.06 | 72.4±0.38 | 72.41±0.06 | 45.5±0.00 | 45.98±0.46 |
| RF-DARTS Zhang et al. (2023) | 91.30±0.36 | 94.27±0.15 | 72.95±0.76 | 72.94±0.81 | 46.40±0.04 | 46.10±0.34 |
| IS-DARTS He et al. (2024) | 91.55±0.00 | 94.36±0.00 | 73.49±0.00 | 73.51±0.00 | 46.37±0.00 | 46.34±0.00 |
| LLMatic Nasir et al. (2024) | - | 94.26±0.13 | - | 71.62±1.73 | - | 45.87±0.96 |
| LAPT-REA Zhou et al. (2025) | - | 94.36 | - | 73.45 | - | - |
| SWNAS* | 91.64±0.11 | 94.41±0.13 | 73.26±0.05 | 73.32±0.10 | 46.22±0.15 | 46.40±0.17 |

*  SWNAS framework without Node Adaptation.

## 7 INTEGRATING SWNAS INTO VARIOUS DARTS-SERIES METHODS

To validate the capability of seamless integration into existing DARTS-family frameworks of SWNAS, we introduced our method to several classical DARTS-series works and tested accuracy improvements on CIFAR-10.

Results in Table 2 confirm that our method provides consistent improvements across different base methods, suggesting its broad applicability and robustness. In particular, the performance gains observed in DARTS-series architectures indicate that our approach not only enhances accuracy but also offers a novel perspective for improving differentiable architecture search frameworks.

Table 2: SWNAS Based on DARTS Variants

| Base Method | ΔAcc(%) |
| --- | --- |
| DARTS- Chu et al. (2020a) | 0.14 |
| PDARTS Chen et al. (2021) | 0.12 |
| PC-DARTS Xu et al. (2019) | 0.19 |
| Fair DARTS Chu et al. (2020b) | 0.15 |
| Average | 0.15 |

# 8 APPLYING SWNAS TO EEG EMOTION RECOGNITION

To evaluate the generalizability of our proposed method, we further applied our model to the task of EEG-based emotion analysis. Specifically, we conducted experiments on the SEED Zheng et al. (2019) dataset for three-class emotion recognition, as well as on the DEAP Koelstra et al. (2011) dataset for arousal, valence, and combined arousal–valence four-class classification. SEED provides multi-session, subject-dependent EEG recordings labeled with three emotional states (positive, neutral, negative), while DEAP contains multimodal physiological data with continuous arousal and valence ratings, commonly discretized into categorical emotion labels for classification tasks.

| Method | SEED (3-class) | DEAP - Arousal | DEAP - Valence | DEAP-A&V (4-class) |
|--------|----------------|----------------|----------------|---------------------|
| baseline | 98.05% | 80.92% | 80.23% | 70.44% |
| RD | 98.11% | 81.78% | 80.47% | 71.56% |
| RD+NA | 98.46% | 82.01% | 80.72% | 71.65% |

baseline: Synergistic Weights without Refined Discretization and Node Adaptation, using traditional rules to discretize (top-2 edges per node).
RD: Refined Discretization
NA: Node Adaptation

Table 3: Performance of SWNAS on SEED & DEAP

During the experiments, we did not alter the originally proposed inference strategy, experimental conditions, or the search space; nonetheless, as shown in Table 3, our method achieves a clear improvement over the baseline models, demonstrating the enhanced representational capability and stronger robustness of the systematic search and reasoning process.

# 9 REASONING PROCESS OF DIFFERENT LLMS

SWNAS proposes a detailed and well-structured strategy that enables consistent performance across various LLMs. By clearly delineating roles, defining reasoning steps, and leveraging both quantitative signals and qualitative architectural insights, our method ensures that different language models can reliably follow the intended logic without relying on model-specific behaviors. This design makes the approach broadly applicable and not tied to any particular LLM.

To validate the generalizability of our strategy, we tested it using Claude-Sonnet-4 and Gemini-2.5-Pro and GPT-5. We present examples of their intermediate reasoning processes (They ran a hidden chain-of-thought reasoning, we could only see the conclusion of their thinking), which reveal their ability to follow our framework and reach meaningful architectural decisions. These examples further highlight how the combination of structured prompts and a fixed strategic framework fosters consistent, interpretable reasoning across different LLM platforms.

Our approach demonstrates robust performance across multiple LLMs, indicating that its effectiveness is independent of specific model implementations. This robustness stems from our structured prompt design and fixed strategic framework, which ensures consistent reasoning patterns across different LLMs. It showcases the power of integrating quantitative metrics with qualitative architectural reasoning—a synergy that enables the discovery of novel design patterns beyond the reach of purely quantitative methods.

## 9.1 OPTIMIZED DISCRETIZATION BY GEMINI-2.5-PRO

*Thinking...*

**Dissecting the User's Intent**

I'm now deep in the deconstruction phase. The prompt's complexity necessitates a meticulous checklist. Key elements are emerging: DAR requirements, sub-network design parameters, and constraints. I'm prioritizing the user's ultimate goal: a functional and efficient "cell" design. The goal is to avoid omissions.

**Formulating Selection Criteria**

I'm solidifying the selection logic. My approach now hinges on applying the rules derived from the prompt. Prioritizing input edge selection using the comparison rules for -2 and -1 sources is next, followed by integrating the internal edge weights. I need to make sure the edge count constraint is precisely satisfied.

**Outlining the Strategy**

I'm laying out the operational blueprint. I've distilled the prompt's essence, forming a detailed requirements list. I'm focusing on the selection rules: edge count, input/internal edge differentiation, and topological constraints. Next I'll structure the data; a table representation seems optimal for edge properties and weights.

**Formulating the Initial Table**

I'm now structuring the data into a usable format. I'm building out tables; one for input and another for internal edges. Sorting and filtering become much more manageable this way. The next step is to begin comparing the input edges, according to the -2 and -1 source designations.

**Evaluating Edge Candidates**

I'm now weighing the input and internal edges. I'm focusing on those criteria, especially the global relative weights, to maintain enough complexity in the structure, while also addressing all the constraints in the selection process. I'm taking care to prioritize the nodes and weights.

**Implementing Edge Selection**

My current task focuses on executing the selection process. I'm prioritizing the top candidates for edges, based on the established rules for inputs, internal connections, and the global weights. The goal now is to hit the target edge count of 8-10.

**Refining the Selections**

I'm now iterating the final edge selections. Reviewing the topology, the goal is to enhance the network's structural complexity. After eliminating the pooling ops, I'm choosing between the internal edges, making sure to hit the 8-10 count.

**Fine-Tuning the Solution**

I'm now refining the edge selection to complete the network, making sure that I have enough inputs and a complex inner structure. Currently I'm selecting the internal edges, making sure I am within the 8-10 edge constraint. The output format and diagram should be easy to understand.

**Summarizing Edge Selections**

I'm now concluding the selection process. Using the global relative weights, and the edge counts, I've confirmed my initial selection meets all criteria. I've ensured I'm within the 8-10 edge limit, and the structural integrity is maintained. The final step is to refine the presentation format.

**Decomposing the Requirements**

I'm now fully immersed in the problem's details, and my plan is solidified. I'll transform the prompt into a precise selection process, focusing on the edge weights and satisfying the structural and count constraints. Data parsing and organization are next on my list.

**Formulating a Plan**

I'm synthesizing the requirements, constraints, and data. The aim is a systematic, step-by-step approach. Parsing, sorting and prioritizing are the immediate goals. Then I will select the best edges to satisfy all requirements and constraints. I'll need to create a visual HTML diagram.

**Formulating a Plan**

I'm now formulating a plan based on the collected criteria and constraints. I'll meticulously categorize the input data, sort based on weight, and implement a decision strategy for edge selection, adhering to all rules. The inclusion of a visual diagram is a key focus.

Of course. I will carefully analyze the provided data and design a sub-network cell based on your specifications. The process will be broken down into clear, logical steps to ensure all constraints are met and the resulting architecture is robust.

REASONING AND STEP-BY-STEP EDGE SELECTION

Here is a detailed breakdown of the selection process, following the rules you've provided.

STEP 1: DATA ORGANIZATION AND INITIAL ANALYSIS

First, I will organize the provided data into three distinct groups for easier comparison: input edges from node -2, input edges from node -1, and internal edges.

**Group 1: Input Edges from Node -2 (Raw Weights)** *Sorted by weight in descending order:*

1. '-2 − > 0': 'sep_conv_3x3', 0.3190
2. '-2 − > 2': 'dil_conv_5x5', 0.3125
3. '-2 − > 1': 'dil_conv_3x3', 0.3087
4. '-2 − > 3': 'dil_conv_3x3', 0.2894

**Group 2: Input Edges from Node -1 (Raw Weights)** *Sorted by weight in descending order:*

1. '-1 − > 3': 'avg_pool_3x3', 0.2713
2. '-1 − > 2': 'skip_connect', 0.2689
3. '-1 − > 0': 'max_pool_3x3', 0.2567
4. '-1 − > 1': 'avg_pool_3x3', 0.2504

**Group 3: Internal Edges (Global Relative Weights)** *Sorted by weight in descending order:*

1. '0 − > 1': 'sep_conv_5x5', 0.1968
2. '0 − > 2': 'sep_conv_5x5', 0.1934
3. '2 − > 3': 'avg_pool_3x3', 0.1934
4. '0 − > 3': 'sep_conv_5x5', 0.1893
5. '1 − > 2': 'sep_conv_3x3', 0.1807
6. '1 − > 3': 'dil_conv_3x3', 0.1744

STEP 2: SELECTION OF INPUT EDGES (4-7 TOTAL)

I will now select the strongest input edges, considering nodes -2 and -1 separately.

- **From Node -2:** The weights are all relatively high and close (0.3190, 0.3125, 0.3087). These represent a strong flow of information from the previous cell's output. To ensure the cell has sufficient input, I will select the **top three** strongest edges.
  - **Select:** '-2 − > 0' ('sep_conv_3x3', 0.3190)
  - **Select:** '-2 − > 2' ('dil_conv_5x5', 0.3125)
  - **Select:** '-2 − > 1' ('dil_conv_3x3', 0.3087)
- **From Node -1:** The weights show a clearer drop-off. The top two are significantly stronger than the bottom two.
  - The strongest edge is '-1 − > 3' ('avg_pool_3x3'). This is a pooling operation.
  - The second strongest is '-1 − > 2' ('skip_connect'). This is an important identity-mapping-like operation.
  - The third strongest is '-1 − > 0' ('max_pool_3x3'). Selecting this would add a second pooling operation from the inputs, which we should avoid per the "avoid a large number of pooling operations" rule.

- Therefore, I will select the **top two** strongest edges from node -1.
- **Select:** '-1 − > 3' ('avg_pool_3x3', 0.2713)
- **Select:** '-1 − > 2' ('skip_connect', 0.2689)

**Input Edge Summary:**

- Total input edges selected: 3 (from -2) + 2 (from -1) = **5 edges**.
- This count is within the required range of 4-7.

STEP 3: SELECTION OF INTERNAL EDGES (TO REACH 8-10 TOTAL)

With 5 edges already selected, we need to choose 3 to 5 internal edges. I will select them based on their global relative weights, while also considering the overall topology and operation types.

- The highest-weighted internal edge is '0 − > 1' ('sep_conv_5x5', 0.1968). This is a strong candidate.
    - **Select:** '0 − > 1' ('sep_conv_5x5')
    - *Current total edges: 6*
- The next two edges, '0 − > 2' ('sep_conv_5x5') and '2 − > 3' ('avg_pool_3x3'), have identical weights (0.1934).
    - '0 − > 2' is a convolution, which adds valuable feature extraction capability.
    - '2 − > 3' is another pooling operation. We have already selected one from the input ('-1 − > 3'). Adding a second pooling operation, especially one with a relatively low global weight, is undesirable.
    - Therefore, I will prioritize the convolutional operation over the pooling one.
    - **Select:** '0 − > 2' ('sep_conv_5x5')
    - *Current total edges: 7*
- The next strongest edge is '0 − > 3' ('sep_conv_5x5', 0.1893). This adds another connection from the important node 0. Selecting this creates a hub-like structure at node 0, which is a common and effective motif.
    - **Select:** '0 − > 3' ('sep_conv_5x5')
    - *Current total edges: 8*
- We are now at the minimum of 8 total edges. We can select up to two more. The next candidates are '1 − > 2' ('sep_conv_3x3', 0.1807) and '1 − > 3' ('dil_conv_3x3', 0.1744).
    - Selecting '1 − > 2' adds another layer of complexity, creating a path from '0 − > 1 − > 2'. This allows for more intricate feature interactions. Node 2 would then be aggregating information from almost all previous nodes ('-2, -1, 0, 1'), making it a powerful feature fusion point.
    - Let's select this edge to increase the inner structure's complexity.
    - **Select:** '1 − > 2' ('sep_conv_3x3')
    - *Current total edges: 9*

This brings us to a total of 9 edges, which is within the 8-10 range. This set provides a good balance of inputs and internal complexity.

STEP 4: FINAL VERIFICATION OF THE ARCHITECTURE

Let's review the selected 9 edges against all constraints.

**Selected Edges:**

1. '-2 − > 0' (sep_conv_3x3)
2. '-2 − > 1' (dil_conv_3x3)
3. '-2 − > 2' (dil_conv_5x5)

4. '-1 − > 2' (skip_connect)

5. '-1 − > 3' (avg_pool_3x3)

6. '0 − > 1' (sep_conv_5x5)

7. '0 − > 2' (sep_conv_5x5)

8. '0 − > 3' (sep_conv_5x5)

9. '1 − > 2' (sep_conv_3x3)

**Constraint Checklist:**

- **Total Edges (8-10):** We have selected **9** edges.

- **Input Edges (4-7):** We have selected **5** input edges.

- **Directed Acyclic Graph (DAG):** All connections flow from a node with a lower index to a node with a higher index (e.g., '-2 − > 0', '0 − > 1', '1 − > 2'). There are no cycles.

- **No Discarded Nodes:**
  - Node 0 receives input from -2.
  - Node 1 receives input from -2 and 0.
  - Node 2 receives input from -2, -1, 0, and 1.
  - Node 3 receives input from -1 and 0.
  - All internal nodes (0, 1, 2, 3) have at least one incoming edge.

- **Sufficient Complexity:** The architecture has multiple inputs, a hub at node 0, and a feature aggregator at node 2. This provides a good level of complexity.

The proposed architecture satisfies all the given requirements.

## 9.2 OPTIMIZED DISCRETIZATION BY CLAUDE-SONNET-4

*Thinking...*

Let me carefully analyze this DARTS neural network architecture search data to design a sub-network.

First, let me understand the structure:

- There are input nodes -2 and -1

- There are internal nodes 0, 1, 2, 3

- Each node has a weight (the number after 'dstnode')

- For each edge, there's an operation and a weight

Let me organize the data:

Node 0 (weight: 0.6324):

- From -2: sep _conv _3x3, 0.3190

- From -1: max _pool _3x3, 0.2567

Node 1 (weight: 0.5965):

- From -2: dil _conv _3x3, 0.3087

- From -1: avg _pool _3x3, 0.2504

- From 0: sep _conv _5x5, 0.1968

Node 2 (weight: 0.6126):

- From -2: dil _conv _5x5, 0.3125

- From -1: skip _connect, 0.2689

- From 0: sep _conv _5x5, 0.1934

- From 1: sep _conv _3x3, 0.1807

Node 3 (weight: 0.6378):

- From -2: dil _conv _3x3, 0.2894

- From -1: avg _pool _3x3, 0.2713

- From 0: sep _conv _5x5, 0.1893

- From 1: dil _conv _3x3, 0.1744

- From 2: avg _pool _3x3, 0.1934

Now I need to select 8-10 edges total, with 4-7 being input edges.

For input edges (from -2 and -1), I should consider edges from each input node separately since they have different information flow strengths.

I'll analyze the input node -2's connections, ranking the edges by their weight. The top connections are sep _conv _3x3 to node 0, followed by dil _conv _5x5 to node 2, then dil _conv _3x3 to node 1, and finally dil _conv _3x3 to node 3. This suggests prioritizing the most strongly weighted connections for network design.

For node -1, the connections show avg _pool _3x3 to node 3 has the highest weight, followed by skip _connect to node 2. The max _pool_3x3 and avg_pool_3x3 connections to nodes 0 and 1 have lower weights, indicating less optimal pathways.

The internal edge weights reveal interesting connection patterns, with sep_conv_5x5 from node 0 to node 1 showing the strongest link at 0.1968.

I'm analyzing potential edge selections, focusing on maintaining a balanced network topology. The goal is creating a directed acyclic graph with strategic node connections that maximize information flow while avoiding redundant pathways.

Considering input edges from nodes -2 and -1, I'll prioritize connections that provide diverse input and prevent isolated nodes.

My current strategy involves selecting top-scoring edges that ensure comprehensive node coverage, with particular attention to maintaining network complexity and input diversity.

I've identified 5 input edges connecting nodes from -2 and -1, which provides balanced input across different network nodes. The selection ensures each node receives meaningful input, with Node 2 benefiting from multiple connection sources.

The next phase involves strategically adding 3-5 internal edges to optimize the network's structural integrity, focusing on maintaining high-performance connections.

I'll select top internal edges like sep_conv_5x5 from 0→1 and 0→2, complemented by an avg_pool_3x3 from 2→3, bringing the total edge count to 8 and enhancing inter-node communication pathways.

Connectivity looks solid, with each node receiving inputs from multiple sources. The network structure ensures comprehensive information flow without isolated nodes.

I'm strategically evaluating pooling operations, considering the current avg_pool_3x3 instances and potential optimization opportunities. The goal is maintaining balanced information aggregation while avoiding redundant pooling mechanisms.

By carefully selecting an additional internal edge connecting node 0 to node 3 using sep_conv_5x5, I've expanded the network's complexity and information transformation capabilities. This approach brings the total edge count to 9, creating a more intricate computational graph.

The connectivity ensures each node receives inputs from multiple sources, enabling rich feature interactions. The network maintains a balanced structure with diverse operations and minimal skip connections.

Looking at this DARTS architecture search data, I'll design a sub-network by carefully selecting edges based on the weights and connectivity requirements.

**Analysis of Input Edges**

**From node -2 (ranked by weight):**

1. -2→0: sep_conv_3x3, 0.3190

2. -2→2: dil_conv_5x5, 0.3125

3. -2→1: dil_conv_3x3, 0.3087

4. -2→3: dil_conv_3x3, 0.2894

**From node -1 (ranked by weight):**

1. -1→3: avg_pool_3x3, 0.2713

2. -1→2: skip_connect, 0.2689

3. -1→0: max_pool_3x3, 0.2567

4. -1→1: avg_pool_3x3, 0.2504

**Analysis of Internal Edges (Global Weights)**

**Ranked by global relative weights:**

1. 0→1: sep_conv_5x5, 0.1968

2. 0→2: sep_conv_5x5, 0.1934

3. 2→3: avg_pool_3x3, 0.1934

4. 0→3: sep_conv_5x5, 0.1893

5. 1→2: sep_conv_3x3, 0.1807

6. 1→3: dil_conv_3x3, 0.1744

**Selected Architecture (9 edges total)**

**Input Edges (5 selected):**

1. -2→0: sep_conv_3x3, 0.3190 (strongest from -2)

2. -2→2: dil_conv_5x5, 0.3125 (second strongest from -2)

3. -2→1: dil_conv_3x3, 0.3087 (third strongest from -2)

4. -1→3: avg_pool_3x3, 0.2713 (strongest from -1)

5. -1→2: skip_connect, 0.2689 (second strongest from -1)

**Internal Edges (4 selected):**

6. 0→1: sep_conv_5x5, 0.1968 (highest internal weight)

7. 0→2: sep_conv_5x5, 0.1934 (second highest internal weight)

8. 2→3: avg_pool_3x3, 0.1934 (tied second highest internal weight)

9. 0→3: sep_conv_5x5, 0.1893 (fourth highest internal weight)

**Architecture Validation**

**Connectivity Check:**

- Node 0: receives from -2

- Node 1: receives from -2, 0

- Node 2: receives from -2, -1, 0

- Node 3: receives from -1, 2, 0

**Design Constraints Met:**

- All nodes included (no discarded nodes)

- Forms valid DAG

- Good input coverage from both -2 and -1

- Complex internal structure with node 0 as a hub

- Edge count within range (9 edges: 5 input + 4 internal)

This architecture provides strong information flow from both input nodes while maintaining good internal connectivity and complexity.

### 9.3 NODE ADAPTATION BY GPT-5

thinking...

We need to discuss how to add a hidden node in the DARTS network. The DARTS cell mentioned by the user contains 4 destination nodes (dst), each with output node weights. Under each destination node, indices -2 and -1 represent input nodes, while 0 and 1 represent internal nodes. We need to suggest adding a hidden node at an appropriate location and ensure its connection method with other nodes conforms to the network topology.

**Propose a hidden node connection scheme**

I suggest adding a new hidden node (node 4), which can receive inputs from nodes with higher output weights or bottleneck nodes, for example, from node 0 and node -1 (considering that node 3 is already connected to -1 and 0, node 2 is highly connected, and node 0 has a high weight). Both node 2 and node 3 may be bottleneck nodes.

**Design a hidden node connection scheme**

I am considering introducing a hidden node H (index 4) that takes inputs from node -1 and node 0 and connects to node 2 and node 3. However, it must be ensured that it does not connect to the final output node to avoid becoming an external output node. In DARTS, output nodes are typically connections of intermediate nodes, but in this cell, destination nodes 0-3 may be outputs.

**Optimize the hidden node connection**

Consider adding a hidden node and using it as an additional source for internal nodes, for example, connecting to dst_node2 and dst_node3. When selecting input nodes, choose those with higher weights or bottleneck nodes (such as node -1 and node 0). However, if inputs are taken from node 2 (which is highly connected), it may form a shortcut path and even potentially cause cycles.

**Adjust the hidden node connection order**

I have decided to use the hidden node as an additional input source for internal nodes, for example, connecting to dst_node2 and dst_node3. The hidden node should receive inputs from nodes with higher weights or bottleneck nodes, possibly selecting node -1 and node 0. However, if connected from node 2, it may create shortcuts but could also lead to cycle issues. Therefore, the hidden node should be placed after node 1 and ensure its index is greater than any source node.

**Propose inserting a hidden node**

The user did not specify the node order, but in a DARTS DAG, nodes are typically arranged sequentially, with edges connecting from smaller nodes to larger nodes. I suggest inserting a hidden node H between node 1 and node 2, connecting node 0 and node 1, and directing its output to node 2 and node 3. This can alleviate bottleneck issues and provide a reasonable connection method.

**Recommend adding hidden node H and modifying connections**

I recommend inserting a hidden node H between node 1 and node 2, newly defined as node h. Node h will receive inputs from node 0 and -1, alleviating bottlenecks and enhancing the functionality of node 2 by providing richer feature inputs. Update dst_node2 and dst_node3 to include this hidden node.

Nice — here's a concrete, self-contained proposal that adds a single hidden (internal) node to the cell and explains exactly where to plug it in, why, and what ops/initialization I recommend.

Quick diagnosis of the existing cell

From the topology you posted:

Node 2 is a hub: dst_node 2 has four sources (-2, -1, 0, 1) — a central aggregator / potential bottleneck.

Node 3 depends only on -1 and 0 (a short, fairly linear path).

Node 0 is fed only from input -2 (a narrow branch).

dst weights are similar (0.5965–0.6378) — no single node dominates, but node2's many incoming edges make it structurally critical.

Goal: add one internal (hidden) node H that is not an external output, to (a) provide an alternative/-parallel path to reduce overly linear flows and (b) relieve / enrich the central aggregator (node2) by pre-processing / fusing useful features.

Concrete topology change (exact patch)

I suggest inserting a hidden node H that collects from earlier useful sources (a compact fusion of -1 and 0) and feeds into the critical aggregators (dst_node 2 and dst_node 3). Keep all existing edges; H is added as an extra source for downstream nodes.

Expressed in the same format you used:

Cell 0 (modified):

dst_node 0, 0.6324:

src_node -2

dst_node 1, 0.5965:

src_node -2

src_node 0

**NEW hidden node H (internal only — NOT part of cell outputs)**

hidden_node H:

src_node -1

src_node 0

dst_node 2, 0.6126:

src_node -2

src_node -1

src_node 0

src_node 1

src_node H # added

dst_node 3, 0.6378:

src_node -1

src_node 0

src_node H # added

Notes:

H is not a cell output. It's only used as a source for downstream nodes (2 and 3).

If your implementation enforces topological ordering by index, insert H before node2 (so indices remain acyclic). Conceptually H is an intermediate node between nodes 0/1 and the downstream aggregators.

Why this choice?

Relieve the hub (node2) — node2 currently aggregates many raw sources. Letting it consume a pre-fused representation from H lets node2 operate on a richer, higher-level feature (H) rather than only on raw/independent streams. That often improves representational depth without increasing output size.

Create a parallel path to node3 — node3 previously had a short path; adding H → node3 provides an alternative pathway that can capture cross-feature interactions that the direct 0 or -1 → 3 paths miss. This reduces overly linear information flow.

Minimal risk to outputs — H isn't exposed as cell output, so it won't change how the cell's output set is constructed. It only enriches internal representations.

Low overhead — a single node with a couple of extra edges is a small structural change but can significantly increase expressivity.

## 10 CELLS SEARCHED ON CIFAR-10

Figure 4 demonstrates the architecture of cells searched by our SWNAS on CIFAR-10. The structures indicate that the cells have been well optimized by SWNAS following the strategies described in section 5.

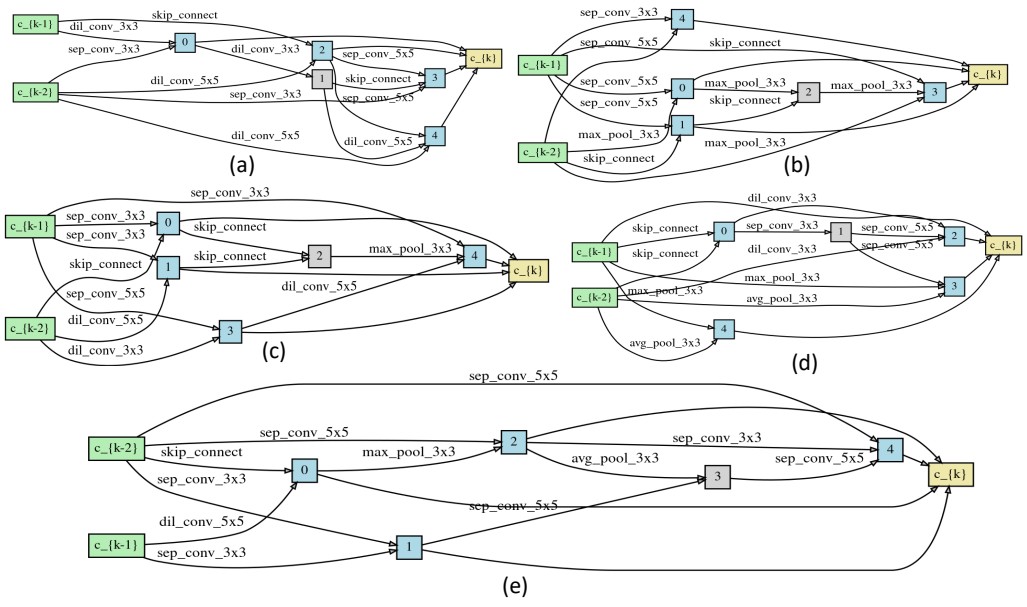

Figure 4: The Cells Searched on CIFAR-10
(a), (c), (e): normal cells.
(b), (d): reduce cells.

## 11 ACTIVATION TIMING OF NODE WEIGHTS MODULE

We observed in preliminary experiments that prematurely introducing the Node Weights Module (NwM) often results in suboptimal final architectures. This phenomenon is primarily attributed to the significant gap between the architecture search phase and the subsequent training phase in Neural Architecture Search (NAS). Specifically, the node weights generated by NwM exert influence on the

internal topology of the cell as well as the global feature representation. However, since NwM is excluded during the actual training stage, architectures optimized in its presence may become poorly aligned with the training environment, thereby degrading performance. To address this issue, we conducted a detailed analysis of the timing of NwM insertion.

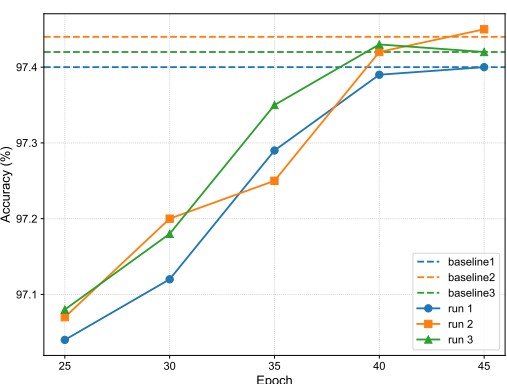

The timing analysis in Figure 5 reveals crucial insights about the NWM's integration strategy. Our experiments show that introducing the module at epoch 40 leaves the final architecture nearly unchanged, indicating that the supernet has already converged. Earlier insertions successfully shift the topology, proving the module can effectively guide the searching process only before convergence while remaining harmless afterward. This finding led to our strategic decision to insert the node-attention module at epoch 45, providing sufficient training time to learn reliable node weights while minimally disturbing the converged architecture.

Figure 5: Different Activation Timing of NWM

## 12 EPOCHS OF NODE ADAPTATION PHASE

After initial discretization, SWNAS employs an LLM-guided Node Adaptation Strategy to further enhance representational capacity. Once adapted, the augmented search space undergoes a brief, targeted training to finalize connections. In this section, we investigate the impact of additional training epochs on network performance. We conducted 2 runs using different discretized architectures. Each run trains the model on CIFAR-10 for 30 epochs and obtains six different architectures by extracting the architecture weights at every 5-epoch interval. We plot a line graph illustrating the relationship between the epochs and network performance. The results in Figure 6 indicate that the performance of the discretized architecture of new

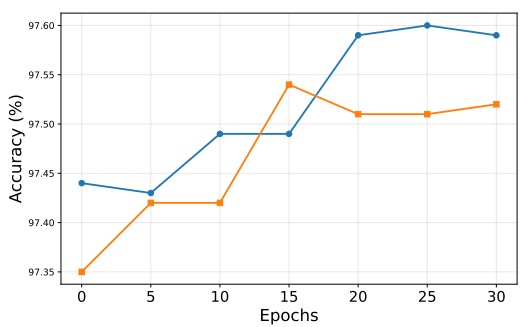

Figure 6: Epochs of Node Adaptation Phase

connections generally converges around 20 epochs. Therefore, conducting 30 epochs in the main experiments can ensure the performance of the final architecture while maintaining computational efficiency.