# OpenReview forum: "Bridging Quantitative Optimization and Qualitative Reasoning: LLM-Enhanced Neural Architecture Search with Synergistic Weights"
_ICLR.cc/2026/Conference — Submitted to ICLR 2026_

### Official Review · Reviewer_8RV1 · 2025-10-28

**Soundness:** 2
**Presentation:** 1
**Contribution:** 2
**Rating:** 2
**Confidence:** 5

**Summary:**

The paper proposes to combine node-level importance scores with edge-level operation weights to produce a joint “synergistic” importance measure for neural architecture search (NAS). The synergistic weights are then fed (in a structured format) to a large language model (LLM) via prompt engineering to produce discretization and node-expansion suggestions. The method achieves competitive results on CIFAR-10 and ImageNet.

**Strengths:**

1. The paper contains a reasonable set of experiments across multiple benchmarks, and the results on CIFAR-10 and ImageNet appear promissing.
2. The authors make an effort to improve reproducibility by including prompts and additional implementation details in the appendix.

**Weaknesses:**

1. The fundamental motivation behind simultaneously considering node and edge importance is not sufficiently explained. The paper would be significantly strengthened by a more in-depth, mechanistic explanation or a motivating example of why this combination leads to better performance beyond the intuitive claim of overcoming "myopic" optimization.
2. The proposed method for learning node importance appears to be a straightforward extension of existing edge-importance learning techniques rather than a genuinely novel methodological contribution.
3. The methodology section lacks clarity and justification for several key design choices. For example, the paper gives the rule “the edge weight α_e is then the maximum A_k” without explaining why taking the maximum is appropriate.
4. The application of the LLM is primarily based on prompt engineering, leveraging its inherent reasoning capability. This paradigm is not novel. The authors should provide a direct comparison (qualitative or quantitative) with recent and highly relevant methods such as [1], [2], [3], and [4] to better position their contribution and demonstrate any unique advantages.

[1] RZ-NAS: Enhancing LLM-guided Neural Architecture Search via Reflective Zero-Cost Strategy
[2] Design Principle Transfer in Neural Architecture Search via Large Language Models
[3] NADER: Neural Architecture Design via Multi-Agent Collaboration
[4] Computation-friendly Graph Neural Network Design by Accumulating Knowledge on Large Language Models

**Questions:**

see Weaknesses.

---

> ### Author Response · Authors · 2025-11-19
>
> Thank you for your insightful comments and constructive criticisms.
>
> weaknesses:
>
> 1.We agree that a deeper, mechanistic explanation is needed. The core motivation is that evaluating edges in isolation ignores the global context of the information flow. In DARTS and its variants, the architecture weight for an operation is used as a coefficient to weightedly sum the operation's output feature map. The node importance learned by our NWM is similarly applied as a multiplicative scalar to its output feature map (Eq. 4). Therefore, the influence of a node and an edge on the final supernet's computation is inherently multiplicative.
>
> 2.We agree that the mechanism for calculating node importance can be implemented using existing attention methods. However, the novelty and contribution of our work do not lie in inventing a new node-weight calculator, but in the conceptual formulation and integration of the Synergistic Weight. The critical insight is that in traditional differentiable NAS, edge weights are only locally comparable within the same source node. Because different nodes have varying feature map scales and semantic levels, comparing an edge from node A to node C against an edge from node B to node C based solely on their α_e values is fundamentally flawed and unreliable. Our key contribution is to introduce node importance as a universal normalizer. We create a globally comparable metric across all edges in the cell.
>
> 3.Thank you for pointing out an error in our description. The A_k in text "calculates operation weights A_k based on AGNAS"(line 222) should be α_e. The calculation of α_e is the mechanism in AGNAS, it's not our proposed new formula.
>
> 4.The role of LLMs in [1][2][3][4] are to understand the search tasks and architectures from text and code levels(only qualitative, understanding architecture and task), but we use LLMs to reason directly with quantitative structural data (synergistic weights, qualitative and quatitative) as mentioned in section 2.4 and table 1. We will add these papers to Related Works.

---

> > ### Comment · Reviewer_8RV1 · 2025-11-26
> >
> > Thank you for your efforts and the detailed clarification. After carefully reviewing the responses, most of my concerns remain unaddressed: the fundamental motivation for simultaneously considering node and edge importance is still not clearly articulated; distinguishing the use of LLMs only from qualitative and quantitative perspectives does not constitute a substantial difference from existing work. After reviewing the authors’ responses and the other reviewers’ comments, I have decided to maintain my score.

---

> ### Author Response · Authors · 2025-11-30
>
> Thank you for your reply.
>
> 1.Our motivation for simultaneously considering node and edge importance comes from this(added to revised version, section 1):
> DARTS series methods often suffer from myopic optimization, primarily due to their reliance on locally computed edge-level weights for connection evaluation. A key underlying issue is the significant variation in output feature map scales across different operations and nodes in the network. While architecture weights in DARTS reflect the relative importance of operations locally (within the same edge or from the same node), they lack global comparability because the magnitude of a weight’s influence depends on the scale of its corresponding input feature map (source node). Consequently, directly comparing these raw weights leads to suboptimal architectures.
>
> 2.We would like to clarify the fundamental distinctions in how LLMs are utilized in SWNAS compared to existing approaches(revised version, section 2.4&2.5):
>
> Interpretability: Some methods treat LLMs as black-box code generators; some methods use LLMs to understand the task from text level. They are both empirical and unreliable. SWNAS uses the LLM as a reasoning module. The full LLM reasoning process in the revised version(supplementary materials, section 8) demonstrated the LLM’s deep understanding of Structure and the clear logic of reasoning. This shifts the role of the LLM from a generator to an interpreter and refiner, leading to more principled and explainable architectural evolution.
>
> LLMs as Interpreters of Quantitative Structural Signals: While some existing methods use architecture performance as a feedback signal for the LLM, this approach remains indirect—the LLM reasons about final results(better/less, still qualitative signals), not the internal optimization process based on weights(quantitative). In contrast, SWNAS employs LLMs to directly interpret and reason over internal quantitative architectural signals. These weights, derived during the gradient-based search, allowing the LLM to act as an integrated reasoner that refines the architecture based on the optimization dynamics themselves, leading to more precise and grounded architectural decisions than what is possible with black-box performance feedback alone.
>
> Tight Coupling Between Optimization and Reasoning
> In existing LLM-NAS approaches, the LLM and the NAS optimizer typically operate in separate, loosely-coupled stages. The LLM functions as an external optimizer or generator, without access to the internal state of the search. In SWNAS, the LLM is deeply integrated into the search loop, directly leveraging the Synergistic Weights as a bridge between gradient-based optimization and high-level structural reasoning. This enables dynamic and context-aware topology adjustments during the search process, a capability absent in prior works.

---

### Official Review · Reviewer_KEzq · 2025-10-28

**Soundness:** 3
**Presentation:** 4
**Contribution:** 2
**Rating:** 6
**Confidence:** 4

**Summary:**

This paper introduces SWNAS (Synergistic Weights Neural Architecture Search), a novel NAS framework that bridges the gap between quantitative optimization and qualitative reasoning. The core idea is to enhance differentiable NAS (specifically, the DARTS family of methods) by integrating LLMs as a reasoning component directly within the search process. The authors propose two main contributions: 1) Synergistic Weights, a metric that combines local edge-level importance with global node-level importance to provide a more holistic signal for architecture evaluation, and 2) LLM-guided Dynamic Search Space Evolution, where an LLM uses these synergistic weights and topological information to perform refined architecture discretization and intelligently expand the search space by adding new nodes. Experiments on image classification tasks and an architectural search space expanded from prior works demonstrate that SWNAS achieves state-of-the-art performance while maintaining computational efficiency.

**Strengths:**

Originality: The core mechanism of integrating quantitative weights and qualitative LLM reasoning is novel.

Methodological Quality: The proposed synergistic weights provide a more robust measure of an edge's importance by considering the significance of its source node, addressing the "myopic optimization" issue in DARTS. The two-phase search process and the LLM-guided node adaptation are designed to avoid known issues in differentiable NAS and to improve architectural exploration.

Clarity: The authors clearly explain the limitations of existing methods and how the newly proposed components are designed to overcome them. The methods are clearly explained overall, aided by visual diagrams and algorithms.

**Weaknesses:**

Limited Scope of the Search Space: The primary experimental validation seems to be performed on a slight expansion of the original DARTS search space. This space is known to be problematic and not representative of many real-world search problems, nor have the resulting architectures been used outside of NAS results reporting. While the use of NAS-Bench-201 (only reported in an appendix) is a good step, this benchmark is also relatively small and limited to the same dataset domains.

Incomparability of Results: The original DARTS search space is expanded in this work, such as by relaxing the constraint of exactly two input edges per node. However, in your main results Tables 2-3, are all of the related frameworks searching in this expanded search space, or only the original one? In the latter case, this is not a fair comparison.

Opaque Computational Cost Analysis: The reported search costs (e.g., 0.16 GPU-days) are misleading as they do not appear to include the computational cost of the LLM inference. The authors mention the LLM process takes "10 minutes", but this is not translated into a standardized metric that can be compared to the GPU-days of other methods. High-end LLMs like Gemini 2.5 Pro require substantial computational resources, and a fair comparison would require an estimation of this cost in terms of equivalent GPU-hours on a reference hardware platform. Without this, the claims of computational efficiency are difficult to verify.

Insufficient Justification for Hyperparameters: The method introduces several new, critical hyperparameters whose values are stated but not thoroughly justified: only the timing of introducing node attention (or NWM, see minor edit below) is empirically explored. Having so many hyperparameters suggests the proposed methods require careful tuning, increasing the true costs of application. Wihtouttheir optimality and sensitivity in question.

**Questions:**

Could the authors provide a more comprehensive breakdown of the computational cost? Specifically, can the LLM inference cost be quantified in a more comparable metric, such as converting all costs to dollar estimates?

The reliance on dated architectural search spaces and tasks is a notable limitation. Have the authors considered how SWNAS might be applied to more modern search spaces? A very relevant search space could be optimizing the organization of different attention-based and state space block types into hybrid architectures for sequence modeling.

The prompts used to guide the LLM (Appendix D) are quite complex and specific. How robust is the method to variations in prompt phrasing? Was significant prompt engineering required to achieve the stable performance across different LLMs reported in Table 5?

Minor edits:
* The NWM seems to be referred to with many different names, such as attention module and NAM. Make sure naming is consistent and clear.
* Use `\citep` whenever the citation is not directly part of the sentence structure, for example "...required by reinforcement learning methods (Zoph & Le, 2017) and...".
* The text in Tables 2-3 and most Figures is too small to read accessibly. Please ensure all text is legible when printed on standard paper size (Letter/A4).

---

> ### Author Response · Authors · 2025-11-19
>
> Thank you for your insightful comments and constructive criticisms.
>
> weaknesses:
>
> 1. In my opinion, the original DARTS search space serve as the standard and foundational framework for the differentiable NAS community. Its primary purpose in research is not to produce architectures for direct deployment in applications. Numerous applied studies in domains like medical image analysis and brain network decomposition have built upon the DARTS framework, tailoring the search space to their specific needs. We did not merely evaluate on a single DARTS variant. As shown in Table 7, we successfully integrated SWNAS into multiple distinct DARTS-family methods (DARTS-, PDARTS, PC-DARTS, FairDARTS). This provides evidence of generalizability beyond the original DARTS constraints.
>
> 2. All compared methods operate within the same original DARTS search space. Our core innovation lies in the discretization strategy. The performance gain is directly attributable to our method's ability to find a better connection pattern, not to a larger or relaxed search space. Besides, the total number of edges in the final discrete architecture (8-10) is kept consistent with the typical output of the original DARTS rule, ensuring a fair comparison in terms of model complexity.
>
> 3. We acknowledge that LLM cost should be transparently quantified. The LLM component operates through a cloud API call(which is easy to use and does not need local resources). The structured data examples and prompts provided to the LLM (detailed in Appendix D) are highly compact, resulting in a very low token count per search. The entire reasoning process for one architecture typically consumes about 1200 x 5(Cells) input tokens and 10000 output tokens. The monetary cost for a single LLM call in our setup is approximately $0.15. We will include this breakdown in the revised manuscript and provide a cost-per-search estimate in the local environment for further transparency.
>
> 4. We agree that hyperparameter sensitivity is a critical concern. Key hyperparameters like number of edges(8-10) were set based on established conventions in the DARTS literature. We will add sensitivity analysis of node adaptation epochs(30 in our experiments) in Apendix.
>
> questions:
>
> 1.weaknesses reply 3
>
> 2.We fully agree that applying SWNAS to modern hybrid architectures is a promising and relevant direction. While our current focus is on CNN-based spaces for fair comparison, the methodology of SWNAS is model-agnostic:
> The Synergistic Weight mechanism can be applied to any differentiable supernet.
> The LLM reasoning strategy can be adapted to reason over block types, connectivity, and depth in transformer-like architectures.
> We are currently exploring SWNAS for attention-based and state-space model search, and will note this as a key future direction in the paper.
>
> 3.The prompts in Appendix D were designed to be strategy-centric and reusable, not heavily tuned per model or dataset. We performed minimal prompt engineering—only ensuring that the LLM follows basic NAS principles. To test robustness, we used the same prompts across all LLMs in Table 5, and all achieved stable performance. Since the context is not too long, we observed no significant performance change and LLMs can fully understand the strategies, suggesting the method does not need significant prompt engineering. We will include a brief discussion on prompt design and stability in the appendix.

---

> > ### Comment · Reviewer_KEzq · 2025-11-24
> >
> > Thank you for the response. While we wait for a paper revision, I would like to discuss Weaknesses 2 and 4. My definition of "original DARTS search space" is the range of valid cells, not the domain before discretization. The valid cells in this search space have exactly 2 inputs per intermediate node (for convolutional cells). Your cells in Figure 10 break this rule, evidence that your methods are granted an unfairly larger search space and thus cannot have their results compared those reported in previous works. Likewise, when you say "Key hyperparameters like number of edges(8-10) were set based on established conventions in the DARTS literature", please cite such claims, particularly using a range across the whole cell, and not exactly 2 edges for each of the 4 intermediate nodes. If you cannot, then to fairly and soundly support your claims, you need to either rerun all related works with a same relaxations granted to your method, or rerun your method to match to correct search space used by others.

---

> > > ### Author Response · Authors · 2025-11-30
> > >
> > > Thank you for your reply. We sincerely apologize for the lack of clarity and precision in our initial manuscript regarding the search space definition.
> > > It is true that our method breaks the strict constraint of the DARTS search space. We would argue that this is not an unfair advantage but rather a core contribution and strength of our approach. Our decision to allow cells with 8-10 edges was deliberate and is justified by the following reasoning:
> > >
> > > Comparability of main experiments: As illustrated in our Figure 3, it is a common phenomenon that some edges within a cell contribute negatively to the overall performance. Our method's global weighting mechanism alleviates this problem by screening more beneficial connections from the search space based on global weights ranking. The standard DARTS framework is unable to mitigate this issue because of its fixed two-inputs-per-node rule. Besides, the node expansion in our method similarly introduces beneficial new connections. Section 5.1 and table 4 also demonstrate that introducing new connections randomly to the architecture would not contribute to the performance. Therefore, a strict comparison with an identical number of edges or params is not entirely meaningful in our main results because it’s hard to keep the same constraints and the ability to break the limitations and introducing more positive connections is what we aim to highlight. Furthermore, our ablation studies in Section 5.1 were designed precisely to ensure the comparability and soundness of our claims. The selection weights assigned by our method exhibit a stronger correlation with the true performance contribution of each edge(higher kendall tau value). This implies that even if we were to enforce the same number of edges as prior works, our method would lead to a better-performing architecture, thereby validating the fairness of our main results.
> > >
> > > Justification for Edge Count Range: Our ablation studies in Section 5.1 were instrumental in determining a suitable edge count. We found that a configuration with 4-7 input edges provided the best balance. This leads to a total of 8-10 edges for the entire cell, ensuring both sufficient information flow from the cell's inputs and a rich, complex internal structure. This range allows us to retain a higher number of positively contributing connections.
> > >
> > > Clarification on "Established Conventions": We acknowledge and apologize for the error in our claim that an edge count of 8-10 is an "established convention" in DARTS literature. This was an inaccurate statement. Our intention was to convey that, during our experimental design, we were mindful of keeping the model complexity (as measured by edge count) reasonably close to that of baseline methods to facilitate a more direct comparison. However, as argued in point (1), enforcing an identical edge count is neither feasible nor desirable, as the primary goal is to showcase the performance improvement enabled by our method's architectural flexibility.
> > >
> > > We have also added experiments of node adaptation epochs in the revised version(supplementary materials section 12).

---

### Official Review · Reviewer_ESRo · 2025-10-31

**Soundness:** 2
**Presentation:** 2
**Contribution:** 2
**Rating:** 2
**Confidence:** 3

**Summary:**

This paper presents an approach for integrating weight-based optimization NAS with LLM-driven reasoning for dynamic search space evolution, thus overcoming key limitations of differentiable NAS. Results are presented on CIFAR-10 and ImageNet.

**Strengths:**

- Novel use of LLMs as high-level reasoning engines that directly interpret architecture node weight data derived from differentiable methods.
- Performance is improved compared to existing methods.

**Weaknesses:**

- Evaluations on Cifar-10 and ImageNet. While the community often uses this benchmarks, I would have liked to see more diverce benchmarks to assess the effectiveness of the approach and the use of an LLM.
- The approach fundamentally relies on a large-scale pretrained LLM model. This might limit the applicability of the approach.
- While performance shows improvement, it is not clear to me if the increase in percentage points is enough to justify the additional complexity of using off-the-shelf large-scale LLMs
- In my opinion the prompt engineering aspect should not be underestimated and should somehow be included in the calculation of the cost. It is a step that would require human oversight.

**Questions:**

- Does the computation include the running of the LLM? Does it run locally or through the cloud? Are the GPU-days reflective of each scenario?
- How do known issues of LLMs, hallucinations, and context size performance degradation, affect the performance of the architecture search.

---

> ### Author Response · Authors · 2025-11-19
>
> Thank you for your insightful comments and constructive criticisms.
>
> weaknesses:
>
> 1.Since most NAS methods were run on CIFAR-10 and ImageNet, applying SWNAS to other benchmarks would be lack of comparability. We evaluated SWNAS on NAS-Bench-201 in Apendix.
>
> 2.We agree that SWNAS relies on LLMs with better reasoning ability, but the LLMs do not need fine-tuned and LLM APIs are cheap and easy to use. So it will not limit the applicability.
>
> 3.As mentioned in the Results(4.2), LLM reasoning process only cost 10 minutes(which is negligible) and it is a one-time step using cloud APIs. The performance improvements are statistically consistent and reproducible, so the gains are achieved without substantial computational trade-offs.
>
> 4.Once the strategy is fixed, it can be reused across searches without further tuning in the same search space. The prompts used in SWNAS are generic and strategy-based, not dataset- or space-specific. We did not include prompt tuning in the cost calculation because it was a one-time development effort, minimizing the need for ongoing human oversight.
>
> questions:
>
> 1.Section 4.2: “The search cost includes the normal search phase, node adaptation phase (0.12+0.03 GPU days) and LLM reasoning process (10 minutes)”
>
> 2.We observed forgetting in DeepSeek-R1. It has a very long thingking process(about 150 seconds). It would ignore some strategies(eg. the edge ranking) in the final output. We did not find hallucinations during experiments. We observed no performance degradation caused by context limits in our experiments. The example data in Apendix D is not too much for LLMs.

---

> > ### Comment · Reviewer_ESRo · 2025-11-27
> >
> > First of all I would like to thank the authors for their response and clarifications. With regards to point 1 I think one of the main objectives of the paper is to show that the addition of LLM provides improvements beyond standard NAS methods. As such comparing on other datasets that are not well known in NAS literature could show the true potential of the approach (with and without the use of LLM) and thus I still believe it is valid. This would also improve the case for point 4 to demonstrate that prompts are indeed generic and reusable beyond standard benchmarks. Furthermore, after also considering the other reviews the methodology and use of LLM in the search process it is still not clear how it substantially improve from prior work. Given these I maintain my initial score.

---

> > > ### Author Response · Authors · 2025-11-30
> > >
> > > Thank you for your reply.
> > > 1.We have conducted additional experiments on EEG-based emotion analysis using the SEED and DEAP datasets. As shown in supplementary materials section 8, our method achieves consistent performance gains across all experimental settings. Specifically, on SEED for 3-class emotion recognition, our approach improves from 98.05% (baseline) to 98.46%. On the DEAP dataset, we observe similar improvements for arousal classification (80.92% → 82.01%), valence classification (80.23% → 80.72%), and the more challenging 4-class arousal-valence task (70.44% → 71.65%).
> > > These results demonstrate that our method maintains its effectiveness beyond conventional NAS benchmarks and exhibits strong generalization capability to different domains and task complexities. The systematic search and reasoning process indeed provides measurable improvements over traditional approaches, validating the core contribution of our paper.
> > >
> > > 2.The key advancement of our work lies in how we integrate it to solve a fundamental limitation in differentiable NAS that prior works could not address.
> > > Prior LLM-based NAS methods use the LLM as an external, separate optimizer or code generator. They operate in a black-box manner, relying on performance feedback or experience, which remains disconnected from the internal quantitative signals. Our method achieves a deep integration where the LLM acts as a Reasoning Engine that directly interprets quantitative architectural signals. This is a paradigm shift, and the substantial improvements come from two core innovations:
> > >
> > > Bridging the Optimization-Reasoning Gap with a Novel Interface: The primary challenge we identify is the inability of gradient-based weights to capture global architectural properties (e.g., structural completeness, balanced information flow). Our proposed Synergistic Weights serve as a novel, information-rich interface between the supernet's optimization process and the LLM's reasoning capability. This allows the LLM to reason about the architecture holistically, which is impossible with local, myopic edge weights used in prior DARTS variants.
> > >
> > > Enabling Topology Evolution Beyond Fixed Rules: Previous methods are confined to pre-defined supernet topologies and rule-based discretization (e.g., "select top-2 edges per node"). Our LLM-guided refinement transcends these fixed rules. By reasoning over the Synergistic Weights, the LLM can perform qualitative discretization and suggest principled node expansions, actively evolving the search space to discover innovative topologies that are not pre-defined. This is a direct and substantial improvement over the static search spaces of all prior DARTS-family works.
> > >
> > > The substantial improvement are validated in section 4.3 and supplementary materials section 7. The SWNAS module consistently improving performance by 0.16-0.19% across frameworks(DARTS and its variants).
> > >
> > > We have also attached the complete LLM reasoning process in supplementary materials section 9. The structured reasoning step demonstrates the systematic design and deep LLM integration of our method.

---

### Official Review · Reviewer_bLjn · 2025-11-01

**Soundness:** 2
**Presentation:** 3
**Contribution:** 2
**Rating:** 2
**Confidence:** 4

**Summary:**

This paper proposes SWNAS, a Neural Architecture Search (NAS) framework that integrates quantitative optimization with qualitative reasoning by introducing two key innovations: Synergistic Weights, which combine edge and node importance for globally-aware architecture evaluation, and LLM-guided dynamic search space evolution, which enables adaptive topology refinement beyond fixed constraints. The method is evaluated on CIFAR-10 (2.33% error) and ImageNet (23.9% top-1 error) while maintaining computational efficiency, and its modular design allows seamless integration into existing DARTS-based methods.

**Strengths:**

++ The experimental evaluation is thorough, with strong results on standard benchmarks (CIFAR-10, ImageNet) and ablation studies validating the contribution of each component. The modular integration into multiple DARTS variants demonstrates robustness and practical applicability.

++ The paper introduces a combination of differentiable NAS with LLM-based reasoning. The concept of Synergistic Weights to address local optimization limitations and the direct use of LLMs for architectural discretization and expansion represent a departure from traditional rule-based or purely optimization-driven NAS approaches.

**Weaknesses:**

-- The method heavily depends on LLMs for discretization and node expansion, but it is unclear whether these models possess the necessary architectural or domain-specific knowledge from pre-training. This may lead to suboptimal or uninterpretable structural choices, as LLMs are not inherently trained for neural architecture design. The reasoning examples in the appendix, while illustrative, do not fully justify the generalization of this approach. Moreover, the interpretability of LLM decisions remains limited, which could hinder adoption in critical applications.

-- While the paper argues that LLMs enable "qualitative reasoning," the prompts and strategies used (e.g., avoiding excessive pooling) are heuristic and could be implemented without LLMs. The added value of LLMs over structured algorithms or expert-defined rules is not convincingly demonstrated, and the risk of LLM-generated artifacts or biases is not addressed.

**Questions:**

1. Experiments are primarily conducted on vision tasks (CIFAR-10, ImageNet) and DARTS-based search spaces. The generalizability to non-vision domains or more complex architectures (e.g., transformers) is not explored. Moreover, can we use this method to refine the architecture of LLM?

2. Can this method adapt to other search spaces? Since the interpretability of LLM decisions remains limited, its reasoning may diverge from optimal architectural principles.

---

> ### Author Response · Authors · 2025-11-19
>
> Thank you for your insightful comments and constructive criticisms.
>
> Weaknesses:
>
> 1.We agree that LLMs are not inherently trained for neural architecture design, but LLMs are not used as domain experts, but as structured reasoning engines that operate on architectural signals. The prompts (Appendix D) explicitly encode NAS-specific strategies—such as maintaining cell width, and ensuring DAG connectivity. We verified the effectiveness of our proposed strategies in Ablation Study.
>
> 2.We also agree that the rules are heuristic, but the combination of multiple constraints—integrity of architectures(make sure it’s a DAG), global weight ranking, cell width control, skip-connect distribution, and dynamic node expansion—creates a complex optimization that is difficult to realize by static heuristic algorithms. The LLMs can effectively balance these factors while remaining flexibility by reasoning.
>
> Questions:
>
> The core methodology of SWNAS is fundamentally generalizable. The principles of Synergistic Weights for global importance evaluation and reasoning for topology refinement are applicable to any neural architecture search space that involves selecting operations and connections within a graph structure, including transformer. But applying SWNAS to a new search space requires the design of strategies that incorporate the domain-specific structural priors and constraints of that search space. To provide a rigorous and focused evaluation, we concentrated our experiments on the well-established and widely-understood DARTS search space.
>
> The LLM reasons following the strategies we proposed in our method. The interpretability of LLM is ensured by structured prompts and constraints, which are evaluated in Ablation Study.

---

> > ### Comment · Reviewer_bLjn · 2025-11-26
> > **Reply to Authors' Rebuttal**
> >
> > Thank you for your response. However, my concern regarding the underlying motivation remains unaddressed: Given that neural network architectures are fundamentally represented as DAGs, and considering the inherent difficulty in accurately describing such complex structures through natural language alone, I am concerned that LLMs may struggle to comprehend the conceptual meaning and functional significance of DAGs without specialized training in this domain.
> >
> > A further concern is the risk of data leakage, wherein knowledge of architecture performance could be implicitly contained within the training corpus. To conclusively demonstrate the inductive reasoning ability of LLMs, it is necessary to employ an abstracted representation of model architectures. This involves substituting all operation identifiers with generic placeholders (e.g., A, B, C) and utilizing natural language to convey solely the DAG's connectivity and associated performance metrics. Such a methodology would enable a valid assessment of an LLM's capacity to identify high-performing architectures from structural descriptions only.

---

> > > ### Author Response · Authors · 2025-11-30
> > >
> > > Thank you for your response. We have attached full LLM reasoning process in the revised version(supplementary materials section 9). It addresses both concerns regarding DAG comprehension and data leakage.
> > >
> > > 1. Demonstrated Understanding of DAG Structure: The step-by-step reasoning shows the LLM does not only process natural language descriptions but operates on an abstract structural representation. It explicitly:
> > > - Groups edges by type (Input from -2, Input from -1, Internal).
> > > - Tracks topological constraints by ensuring all selections flow from lower to higher-indexed nodes ("All connections flow from a node with a lower index to a node with a higher index... There are no cycles").
> > > - Manages node connectivity by checking that "All internal nodes (0, 1, 2, 3) have at least one incoming edge," demonstrating an understanding of the DAG's connectivity requirements beyond a simple edge list.
> > >
> > > 2. Reliance on Structure, Not Operation Semantics: The reasoning process confirms that the LLM's inductive bias comes purely from the weight-based DAG topology, not from prior knowledge of operations. As stated in the trace:
> > > "I will select them based on their global relative weights, while also considering the overall topology..."
> > > This is evidenced when the LLM ranks and selects edges solely by weight (e.g., selecting the top 3 from node -2, top 2 from node -1). Treats operations as generic types when making structural decisions.
> > > We have also tried using A,B,C to replace operations and we got similar reasoning process. The results are easy to verify.

---

### Meta-Review · Area_Chair_jPeg · 2026-01-12

**Summary:**

The reviewers’ concerns are mainly about the possibility that the method may lead to suboptimal or uninterpretable structural choices, the interpretability of LLM decisions, the DAG structure, data leakage, and the need for more diverse benchmarks. They also raise concerns about the novelty of node-importance learning, the limited scope of the search space, the incomparability of results, opaque computational cost analysis, and insufficient justification for hyperparameters.

The authors’ rebuttal has addressed most of the concerns. However, the issues of suboptimal structural choices, method novelty, the limited scope of the search space, and the justification of hyperparameters have only been addressed to a limited extent. The proposed method for learning node importance appears to be a straightforward extension of existing edge-importance learning techniques rather than a genuinely novel methodological contribution.

The rebuttal is sometimes difficult to follow, unstructured and vague, e.g., “Weaknesses Reply 3.”

**Reviewer Concerns:**

The authors’ rebuttal has addressed most of the concerns. However, the issues that the method may lead to suboptimal structural choices and that it represents only an incremental extension of existing edge-importance learning techniques are still outstanding. The limited scope of the search space and the justification of hyperparameters have been partly addressed but remain unsatisfactory.

**Reviewer Scores:**

I believe most reviewers will maintain the original score; a few might increase or decrease it, but it won't significantly affect the average.

---

### Decision · Program_Chairs · 2026-01-26

Reject